# High spatio-temporal velocity variations driven by water input at a Greenlandic tidewater glacier

Armin Dachauer<sup>1</sup>, Andrea Kneib-Walter<sup>1</sup>, Dominik Gräff<sup>2</sup>, and Andreas Vieli<sup>1</sup>

**Correspondence:** Armin Dachauer (armin.dachauer@geo.uzh.ch)

Abstract. Ice flow controls the ice discharge at tidewater outlet glaciers and is, together with frontal ablation, a key contributor to the mass loss of the Greenland ice sheet. While annual glacier velocity variations of tidewater glaciers are well studied using satellite-derived data, research on their small-scale, short-term speed variations, ranging from sub-diurnal to multi-day scales, remains limited. We deployed a terrestrial radar interferometer, operating at a temporal resolution of one minute and a spatial resolution of a few meters, to investigate small-scale ice flow variations at the terminus of Eqalorutsit Kangilliit Sermiat, a tidewater outlet glacier in South Greenland. We observed clear diurnal and multi-day ice flow speed variations and link these to a high sensitivity of the glacier system to additional freshwater input. This water originates from different sources, such as enhanced surface melt during warm periods or sudden drainage events from subglacial or ice-marginal lakes. The amplitudes of diurnal velocity fluctuations remain remarkably consistent throughout the 6 km long terminus area, but their spatial evolution shows clear local variability. Spatio-temporal analysis of velocity map time-series revealed a general downstream propagation of diurnal velocity variations. However, on days characterized by particularly high ice flow speeds (multi-day speed-up events), these variations start at the terminus propagating upstream in a distinct block-wise pattern, connected to major rifts in the terminus area. Our high spatio-temporal resolution data underline the complex influence of water input and basal hydrology on the dynamics of tidewater glaciers.

#### 15 1 Introduction

Over the past few decades, global warming has caused the Greenland ice sheet (GrIS) to lose mass at an increasing rate (Murray et al., 2015a; Mankoff et al., 2021), making it the largest contributor to global sea level rise (Bamber et al., 2019; Shepherd et al., 2020). About two-thirds of this mass loss between 1972 and 2018 is attributed to glacier dynamics of marine-terminating glaciers (Mouginot et al., 2019). Our current understanding of the impact of environmental forcings, such as melt water input, on flow dynamics of tidewater glaciers is limited. Several studies suggest that warmer temperatures in the future, and the associated increase in surface melt, lead to enhanced mass loss through accelerated dynamic thinning (e.g. Parizek and Alley, 2004; Pritchard et al., 2009). However, other studies indicate that this relationship is not universally applicable, since not the average melt but rather the melt variability influences increased flow velocities (Schoof, 2010). Consequently, it is crucial

<sup>&</sup>lt;sup>1</sup>Department of Geography, University of Zurich, Zurich, Switzerland

<sup>&</sup>lt;sup>2</sup>Department of Earth and Space Sciences, University of Washington, Seattle, USA

50

to understand the small-scale flow variations at the trunks of the ocean-terminating outlet glaciers, with short-term velocity observations being a key constraint for such analysis.

Annual or seasonal variations in ice flow dynamics of the GrIS have been well studied, mostly using satellite-based observations (e.g. Joughin, 2022; Gardner, A. S. et al., 2023). Satellite remote sensing techniques, such as Synthetic Aperture Radar (SAR) and optical imagery, have proven valuable for large-scale, long-term monitoring of ice velocity and its response to a warming climate, in particular in large and inaccessible regions such as Greenland (e.g. Joughin et al., 2010; Tsai et al., 2019; Joughin, 2022). However, a few studies show that tidewater glaciers also undergo short-term velocity variations at a sub-diurnal to multi-day scale (e.g. Meier et al., 1994; Vieli et al., 2004; Kneib-Walter et al., 2023; Sugiyama et al., 2025). They highlight the impact of diurnal meltwater input or tidal forcing on small-scale speed fluctuations, where additional water enters the glacier system leading to a higher subglacial water pressure, which enhances basal sliding and results in larger ice flow velocities (Iken, 1981; Stevens et al., 2022a). Throughout all timescales, the evolution of the subglacial drainage system has a large impact on the ice dynamics (Nienow et al., 2017). Despite multiple efforts to investigate these rapidly changing systems (e.g. Chandler et al., 2013; How et al., 2017; Doyle et al., 2018), they remain poorly understood due to the challenges involved in their investigation. This particularly applies for tidewater glacier tongues where basal water pressures are generally rather high due to a bed well below sea level.

Improving our process understanding of detailed dynamics at the ice-ocean boundary requires short-term and high-resolution observations. However, the multi-day revisit frequency of satellite data limits their ability to capture short-term glacier dynamics, highlighting the continued importance of in-situ observations (Fahrner et al., 2024). Field studies on short-term ice flow dynamics and their relation to subglacial conditions and melt water input largely focus on land-terminating glaciers, which are often relatively small and slow flowing, and thereby much easier to access (Chandler et al., 2013; Cowton et al., 2013). On tidewater glaciers in Greenland, particularly near the terminus, such studies are rare. A common method to determine the ice flow velocity at sub-seasonal or even diurnal timescales are fixed GPS sensors, typically installed along the centreline (e.g. Podrasky et al., 2012; Stevens et al., 2022a; Sugiyama et al., 2025; Wehrlé et al., 2024, preprint). However, deploying and maintaining a GPS network in glacial environments is logistically challenging and can pose significant risks to field personnel, especially on heavily crevassed termini of tidewater glaciers. Additionally, GPS sensors only provide velocity data for single points on the glacier, which limits our spatial understanding of the dynamical processes. Ground-based photogrammetry methods using time-lapse cameras can serve as an alternative (Ahn and Box, 2010; Murray et al., 2015b), but often struggle to produce sub-daily velocity fields, since a considerable displacement is needed to overcome the measurement uncertainty. Uncrewed Aerial Vehicles (UAVs) are limited by their range and the periodic nature of flight missions and therefore cannot deliver continuous data (Ryan et al., 2015; Jouvet et al., 2018). To date, no such studies have been conducted on tidewater glaciers in South Greenland. At Eqalorutsit Kangilliit Sermiat (EKaS), our target glacier, previous studies have examined the glacier's outline change (Weidick, 2009), investigated calving-driven fjord dynamics (Gräff et al., 2025), and documented a substantial subglacial winter meltwater discharge (Hansen et al., 2025). However, none of these works have addressed glacier flow velocity at the site.

To address these limitations, we deployed a terrestrial radar interferometer (TRI) to investigate short-term and small-scale ice flow variations over the terminus area of a major outlet tidewater glacier in South Greenland. The TRI continuously operated over two separate two-week periods in the summers 2023 and 2024 from a fixed position on land, covering the entire terminus area at the lowest 6 km of the glacier with a temporal resolution of one minute and a spatial resolution of a few meters. We further monitored external factors such as meteorological conditions, tides, calving activity, and surface ice conditions in the fjord to investigate their potential influence on ice dynamics.

## 2 Data and methods

## 2.1 Field site

Eqalorutsit Kangilliit Sermiat (61.36° N, 45.76° W), hereafter EKaS (also locally called Qajuuttap Sermia), is an ocean-terminating outlet glacier in South Greenland (Fig. 1). It flows into the Sermilik fjord, approximately 27 km north-west of the international airport Narsarsuaq. EKaS drains a large basin of the inland ice of the GrIS, extending to the East-West ice divide and covering an area of about 5800 km² (0.3 % of the GrIS). The glacier's terminus has advanced over the past few decades (Weidick, 2009), despite being situated in a region where glaciers are generally experiencing substantial thinning and retreat. EKaS has an approximately 3 km wide and 360 m high calving front, which is grounded at a water depth of 280 m (Rosier, 2025). The flow speed of the EKaS terminus varies between 5 m/d at the end of the melt season and 12 m/d at the beginning of the melt season (Gardner, A. S. et al., 2023). This results in an annual ice discharge of approximately 3.1 km³/yr (Mankoff et al., 2020a) in the recent decade.

# 75 2.2 Field data collection

#### 2.2.1 Radar

A terrestrial radar interferometer (TRI) was deployed twice for a two-week period in August 2023 and July 2024, respectively, to continuously measure ice flow velocity. The instrument was positioned on solid bedrock atop an opposing hill, 496 m above sea level and three kilometres from the calving front (Fig. 1). The TRI developed by GAMMA Remote Sensing, is a real-aperture radar interferometer with one transmitting antenna and two receiving antennas, operating at a wavelength of  $\lambda$  = 17.4 mm (Ku-band, 17.2 GHz). The antennas are rotating along the vertical axis on a precision astronomical mount. The range resolution is about 0.75 m, while the azimuth resolution is 0.4°, which corresponds to 6.9 m at a slant range of 1 km (Werner et al., 2008a, b). Whereas the TRI scan has a maximum range of 16 km, our specific location of the TRI system and the glacier topography enable seamless, comprehensive coverage within the first 6 km of the terminus area. In this study, the data acquisition, which is daylight- and weather-independent, was repeated at 1-minute intervals, allowing to receive almost continuous line-of-sight (LOS) velocity and DEM values.

**Figure 1.** Overview Sentinel-2 image (Copernicus Data Space Ecosystem, 2025) showing the terminus region of the tidewater glacier Eqalorutsit Kangilliit Sermiat (EKaS). Superimposed is the mean line-of-sight velocity field derived from the terrestrial radar interferometer (TRI) for the period 3-15 August 2023. The sensor locations of TRI, time-lapse cameras, automatic weather station (AWS) and the tide-gauge are marked. The two inset graphs on the right illustrate the position of the EKaS glacier system within the Greenland ice sheet.

Calving events exceeding a volume of 5,000 m<sup>3</sup> were detected using DEM subtraction from sequential DEMs generated by the interferogram between two receiving antennas of the TRI. The method is thoroughly described in Walter et al. (2020) and Kneib-Walter et al. (2021).

# 90 2.2.2 Weather station

In 2023, we installed a basic automatic weather station (AWS) near the shore east of the terminus of EKaS (data used in Fig.2b). One year later, a weather station was deployed next to the TRI, on the hill on the opposite side of the glacier terminus, 500 m above sea level (data used in Fig.3b). Both weather stations were measuring air temperature, relative humidity, incoming solar radiation, and precipitation at a 30 min interval. Wind speed and wind direction were taken from the Mittafik Airport weather station in Narsarsuaq (Drost Jensen, 2023). This weather station, operated by the Danish Meteorological Institute (DMI), is located about 20 km from the front of EKaS, in a valley that is aligned parallel to the flow direction of EKaS. Since different weather stations were used for the air temperature records in the two years, we compared them with the AWS in Narsarsuaq to ensure that they captured the regional temperature trends rather than local effects (e.g. inversion) influencing the signal (Figs. A1 and A2).

# 100 2.2.3 Time-lapse camera

A time-lapse camera was installed on the hill opposite the calving front (Fig. 1) and was running year-round from July 2022 onwards. The camera took pictures of the calving front at intervals between two minutes during the summer field campaigns and 20 min to one hour for the rest of the year. This allowed us to classify the evolution of the ice mélange extents and subglacial plume extents in classes of none, small, medium and large. Both variables were extracted twice a day in the time-series of Figs. 2d and 3d. A second camera on the hill east of the terminus, running at the same intervals, allowed to estimate the rough timing of a major subglacial lake drainage event L1 from the western tributary of EKaS.

# 2.2.4 Lake drainage events

Besides the time-lapse data, we used satellite imagery from Sentinel-1 and Sentinel-2 (Copernicus Data Space Ecosystem, 2025) as well as data from the Arctic-DEM (Porter et al., 2023) to constrain the timing, extent and the approximate volume of the subglacial lake drainage event L1 mentioned above and an additional marginal lake drainage event L2 on the orographic right side of EKaS, 20 km upstream of the terminus. For the latter, discharge volumes were provided by Dømgaard et al. (2024), based on satellite and airborne altimetry data. The locations of the two lakes are shown in Fig. A3.

# 2.2.5 Tide gauging

A pressure sensor was installed in the north-eastern end of the fjord about 4 km from the calving front (Fig. 1) to monitor the tides. We installed a RBR-duet<sup>3</sup> pressure-temperature sensor in a steal-pipe drilled onto bedrock at the shore in July 2022. It continuously sampled the water level at a 4 s interval until end of July 2024.

#### 2.3 Data Processing

# 2.3.1 Processing of TRI velocity

Ice flow velocities were derived from the TRI phase records using the GAMMA software stack and following an established workflow (Caduff et al., 2015; Kneib-Walter et al., 2023; Wehrlé et al., 2021). The TRI system sends out radar signals from a transmitter antenna and captures them using two receiver antennas. Temporal interferometry was obtained by analysing the phase signal recorded by a single receiver antenna (in this case, the upper antenna) at consecutive acquisition time intervals of 1 minute. To minimize atmospheric noise in measurements caused by variations in air turbulence and humidity (Goldstein, 1995), interferograms collected within 30 minutes were stacked, resulting in a final temporal resolution of 30 minutes. The phase signal was then unwrapped to derive the LOS displacement, which has a sensitivity of less than 1 mm. Finally, the displacement data were used to calculate LOS velocity maps that allow for investigation of spatial and temporal flow speed variability (Werner et al., 2008b).




# 2.3.2 Temporal analysis of TRI velocity

The temporal variability of the velocity during both two-week campaigns was mainly analysed using the values along the centreline of the mapped area (Fig. 1). Averaged velocity estimates close to the front can be affected by boundary effects such as from missing ice due to calving events happening within the 30 minutes between two consecutive velocity maps. Additionally, atmospheric noise increases with distance from the calving front due to longer travel times of the radar beams through the atmosphere. Therefore, the averaged centreline velocities were calculated using data for the first two kilometres along the centreline, where data quality is highest, but excluding the initial 100 meters. The according stretch is labelled as "centreline part" in Fig. 1.

Acceleration maps were derived from smoothed velocity maps as follows: For each pixel in the velocity map, a temporal smoothing was applied. Therefore, a 30-min interval time-series covering the entire two-week period was processed for each pixel using a Butterworth low-pass filter with a cut-off period of 3 hours (Figs. 2a and 3a). The filtered value for each time-step was subsequently used to update the corresponding pixel in the velocity map. Finally, the gradient between two consecutive smoothed velocity data points was calculated and divided by the time difference, yielding acceleration estimates for every time-step and pixel.

# 3 Results

Velocity time-series for both summer campaigns show diurnal variations superimposed on multi-day speed-up events. We first present the temporal velocity results, structured by the different time scales alongside their potential external influences, followed by the spatial variations.

#### 3.1 Temporal ice flow variability

# 3.1.1 Diurnal velocity variations driven by melt

Both line-of-sight (LOS) ice velocity time series from the 2023 and 2024 summer campaigns show clear diurnal fluctuations, which are characterized by an increase in velocity during the day followed by a decrease during night (Figs. 2a and 3a). While this diurnal signal is consistently observed in the 2024 dataset, it is slightly less clear in 2023, particularly during the period from August 10-11. The diurnal acceleration generally starts in the morning around 9:00 local Greenlandic time (UTC-2 in 2023, UTC-1 in 2024) and the velocities peak in the evening around 20:00. However, the exact timing can vary up to six hours between individual days. An average diurnal velocity fluctuation, excluding periods that are heavily influenced by multi-day speed-up events, shows a peak-to-peak amplitude of about 0.5 m/d in both years, which corresponds to about 7 % of the average speed.

The diurnal variation of the flow velocity is for both summers clearly correlated with the air temperature signal measured at a nearby weather station, which acts as a proxy for surface ice melt (Figs. 2b and 3b). Cross-correlation analysis of the velocity and air temperature time-series reveals a peak correlation of 0.6 at a lag of 4 hours, indicating a moderately positive



**Figure 2.** a) Line-of-sight (LOS) mean centreline velocity of the summer campaign 2023 (gold) together with its low-pass filtered representation (green). b) Air temperature, c) relative humidity (RH) and wind records from nearby weather stations. d) Categorical time-series of ice mélange and plume extent are retrieved from time-lapse imagery, and calving events (>5,000 m<sup>3</sup>) detected by TRI. e) Tidal amplitude from tide-gauge.

relationship throughout the entire 2-week field period. In other words, the ice flow speed reaches its daily maximum 4 hours after the temperature peak. For the days from July 16, 2024, onward, the correlation becomes even stronger, reaching a cross-correlation of 0.8, again at a lag of 4 hours. However, before July 16, no correlation between flow speed and air temperature can be detected, with a cross-correlation value 

**Figure 3.** a) Line-of-sight (LOS) mean centreline velocity of the summer campaign 2024 (gold) together with its low-pass filtered representation (green). b) Air temperature and precipitation, as well as c) relative humidity (RH) and wind records from nearby weather stations. d) Categorical time-series of ice mélange and plume extent are retrieved from time-lapse imagery, and calving events (>5,000 m<sup>3</sup>) detected by TRI. e) Tidal amplitude from tide-gauge.

throughout the 2024 measuring period, whereas it fluctuated in size in 2023. On a multi-day scale no clear link to velocity variations was found for either the ice mélange or the plume extent. Further, the diurnal ice velocity variations are also not directly correlated to the tidal signal (Figs. 2e and 3e), which is dominated by a shorter periodicity of 12.4h instead of 24h (Ross, 1995). In both years and regardless of any lag, the correlation values between the tide and the ice velocity are close to zero.

## 3.1.2 Multi-day speed-up events

During both field campaigns multi-day variations in velocity are superimposed onto the diurnal changes. In the 2024 campaign, a very distinct speed-up event occurred between July 19-21 (Fig. 3a), starting with a sudden velocity increase of 2 m/d (28 % of the average speed) within 13 hours, followed by a large diurnal fluctuation peak-to-peak amplitude of 1 m/d (14 %),






before dropping back down to pre-event velocity levels. At the same time, the air temperature rose drastically to almost  $20\,^{\circ}$ C, accompanied by a noticeable shift in the prevailing wind direction from south-west to north-east (Fig. 3c). This shift comes along with a low relative humidity of  $30\text{-}50\,\%$ , which indicates a warm, foehn-like wind descending from the ice sheet. Later, another one-day speed-up event occurred on July 25, with a large velocity increase of  $1.2\,\text{m/d}$  ( $17\,\%$ ), followed by a sudden speed drop of  $1.8\,\text{m/d}$  ( $26\,\%$ ), mostly driven by a large local velocity change signal at the front.

In 2023, a distinct velocity peak is observed between August 7-10 (Fig. 2a). During this period, there is a continuous velocity increase of in total 1.5 m/d (27%), followed by a fast velocity decrease of 1.7 m/d (30%) within 13 hours. Then, another continuous increase of similar magnitude over a period of at least 4-days occurred. The weather, however, remained rather stable, with no substantial additional water inputs such as rainfall, variable wind conditions, or foehn events (Fig. 2c). So, no direct relation to meteorological conditions is apparent for this multi-day speed-up event. Similarly, in the initial phase of the 2024 time-series, a clear slowdown over 3-4 days can be observed on top of some diurnal variations, while air temperatures remain rather low or even slightly increase. This is also reflected in a weak cross-correlation between velocity and temperature. For both years, these longer-term velocity speed-ups match in timing to subglacial and marginal lake drainage events, respectively.

## 3.1.3 Lake drainage events

The slowdown in the first 3-4 days of the 2024 dataset cannot solely be explained by surface melt, but coincides with the cessation of enhanced discharge from a large lake drainage event that may influence the ice flow. A major subglacial lake drainage event (labelled L1), expected to release approximately 100-300 million m<sup>3</sup> of freshwater, has been detected in the time-lapse imagery from below the western tributary about 3 km upstream of the glacier front. The onset of the lake drainage event started already on July 4, a few days before the start of our velocity record, and lasted until July 15.

Having deployed our TRI from July 12 onwards, we only managed to capture the ending tail signal from this lake drainage event, but the timing corresponds well to the phase of slowdown in flow speed and weak cross-correlation between air temperature and ice flow speed. Note, the time-lapse cameras also recorded this large subglacial lake drainage L1 in summer 2023. But this discharge event occurred between July 28 and August 3, stopping just as we began our field campaign, and hence we could not detect any influence in our TRI-velocity record.

For the period of the observed main velocity speed-up event in the summer 2023 data, we observed another marginal lake drainage event (labelled L2), that may have provided a large volume of freshwater input to the glacier system. Satellite imagery from Sentinel-2 (Copernicus Data Space Ecosystem, 2025) showed that the lake, which is located 20 km upstream at the orographic right margin of the main glacier, drained between August 7-9 (Fig. A3). Sentinel-1 data (not shown here) indicates a half empty lake on the morning of August 8 and thus confirms ongoing drainage (Copernicus Data Space Ecosystem, 2025). Dømgaard et al. (2024) already observed this marginal lake drainage in 2020 and 2022 and used satellite products to estimate an average water level drop of 12 m, which corresponds to about 2 million m³ of released water, considering a lake area of roughly 0.2 km². Given an expected delay due to the large distance to the terminus, the lake drainage date seems to align with the high velocity speed-up event that peaks at the start of August 10.

**Figure 4.** a) Smoothed 2023 LOS velocities along centreline and with time, b) their deviations from the 2-week row average in 2023 along centreline and with time, c) smoothed LOS acceleration along centreline and with time. The centreline spans from 100 m behind the calving front to about 5.5 km upstream. The coloured boxes label the along glacier propagation direction of the acceleration transitions, with green for upstream and orange for downstream. Orange-green striped transitions refer to either no clear propagation direction or a combination of an upstream and a downstream signal. White boxes represent no transition (deceleration only).

# 3.1.4 Calving events


The TRI allowed to detect calving events with subaerial volumes larger than 5,000 m<sup>3</sup> and relate them to our velocity variations. In 2023, a total of 134 events were captured during the 12 day field period, whereof the interquartile range (IQR) shows subaerial calving volumes from 10,000 to 30,000 m<sup>3</sup>, with a median calving volume of about 15,000 m<sup>3</sup>. In 2024 (total of 81 events in 14 days), the IQR of the subaerial calving volumes goes from 8,000 to 60,000 m<sup>3</sup>, with a median calving size of about 30,000 m<sup>3</sup>. In both years, the largest captured calving events had a subaerial volume of about 200,000 - 300,000 m<sup>3</sup>.

Throughout the two summer campaigns, we observe that calving events are generally more prevalent during high-velocity periods (Figs. 2d and 3d). For example, the high-velocity day of July 20, 2024, recorded 9 events, compared to just 2 events on July 14, 2024, a day with relatively low flow velocity. The average flow velocity during calving events is significantly higher compared to periods without calving (p

**Figure 5.** a) Smoothed 2024 LOS velocities along centreline and with time, b) their deviations from the 2-week row average in 2024 along centreline and with time, c) smoothed LOS acceleration along centreline and with time. The centreline spans from 100 m behind the calving front to about 5.5 km upstream. The coloured boxes label the along glacier propagation direction of the acceleration transitions, with green for upstream and orange for downstream. Orange-green striped transitions refer to either no clear propagation direction or a combination of an upstream and a downstream signal.

# 3.2 Spatial ice flow variability

## 3.2.1 Spatially coherent diurnal velocity change amplitude

The temporal patterns, such as the diurnal and the multi-day speed-up events, can generally be observed almost uniformly along the entire 6 km long centreline (Figs. 4 and 5). The velocities are generally increasing towards the terminus as typical for tidewater glaciers (Figs. 4a and 5a). Subtracting the 2-week velocity average for each location indicates that the diurnal velocity fluctuations not only appear along the entire 6 km stretch, but also occur at a similar absolute magnitude along the centreline (Figs. 4b and 5b). This is further supported by comparing the average LOS velocity on three different flow and transverse lines, showing an almost constant velocity difference between these lines, regardless of any fluctuations (Figs. A4 and A5). However, a close inspection of the velocity change and acceleration data (Figs. 4 and 5) shows that the timing of the diurnal velocity peak and minima also varies spatially along the centreline and is further analysed below.





# 3.2.2 Spatial patterns of diurnal velocity change propagation

During high-velocity days, diurnal velocity changes initiate at the terminus and propagate upstream, whereas commonly diurnal accelerations start at locations further upstream and propagate downstream over time. The initiation of acceleration in the morning and deceleration in the evening generally occurs with a lag of about two hours between the terminus and the location 6 km upstream. This is visible in the inclined acceleration/deceleration signal with time on Fig. 4c and Fig. 4c. In nearly half of the observed initiations, the velocity change starts earlier at the upper part, propagating downstream with time (marked as orange boxes in Fig. 5c), whereas upstream propagations, when velocity changes are initiated at the front, only occur in 20 % of the cases (green boxes in same figure). The remaining 30 % labelled as "unclear" show either no clear propagation direction or a combination of an upstream and a downstream signal. Table B1 in the Appendix provides a complete overview of the acceleration transition types for both years.

Representative 30-minutes interval time-series of velocity variation maps illustrate in more detail the spatial patterns during propagation of such flow transitions (Fig. 6). These transitions occur either from acceleration to deceleration in the evening (a and c), or from deceleration to acceleration in the morning (b and d). The more common downstream propagation to the terminus appears spatially smooth and uniform across the entire width of the glacier, apart from the almost stagnant parts beyond the shear margins (Fig. A6). In contrast, on days with high flow velocities, such as for example during the speed-up event around July 20, 2024 (Fig. 5c), the transition signal generally starts at the front and propagates upstream. In these cases, propagation typically exhibits a more 'block-like' spatial pattern (Fig. 6a/b). The outlines of these blocks align with major crevasses or rifts, which are oriented in north-westerly direction due to generally faster flow velocities towards the western margin of the glacier (Fig. A6). The velocities are significantly higher (with p



**Figure 6.** Map sequences of deceleration in the evening propagating a) upstream or c) down-stream, as well as acceleration in the morning propagating b) upstream or d) downstream. Each sequence covers a total period of 1.5 h with a time interval of 30 min (absolute time is given in local Greenlandic time). The background image is a Sentinel-2 acquisition from Aug 7, 2023 (Copernicus Data Space Ecosystem, 2025).

that excluded tidal influences as a cause of diurnal velocity fluctuations (e.g. Davis et al., 2014; Pimentel et al., 2017; Stevens et al., 2022a). In particular, tidewater glaciers that are clearly grounded—such as EKaS—have been found to show no tidal response (Kneib-Walter et al., 2023).

The clear correlation between air temperature, which behaves as a proxy for surface ice melt, and ice velocity at EKaS (Figs. 2 and 3) highlights the impact of surface water production on tidewater glacier dynamics. In other words, increased air temperature during the day drives ice melt and therefore increases the freshwater input to the glacier system. This results in higher subglacial water pressure, followed by reduced resistance to basal sliding and therefore enhanced ice speed (Iken, 1981; Stevens et al., 2022a). On days with precipitation, which only occurred during the 2024 field season, the diurnal signal got weakened both for the temperature and the velocity signal (Fig. 3b). Especially on July 23, 2024, the rainfall partially compensated for the lower melt discharge, resulting in only a small velocity drop during night. However, since the precipitation rate was relatively low, its impact on velocity was small.




The surface water production depends not only on air temperature, but also on other factors such as cloud cover and wind speed. This likely explains the large diurnal variability in velocity amplitude and peak timing, even for days with a similar mean air temperature. Since the surface of EKaS is heavily crevassed (Fig. A7), the generated melt water will quickly enter the glacier system (Andrews et al., 2014), where it influences the basal water pressure. At EKaS, the diurnal velocity maxima occurs about 4 hours after the local temperature peak. This delay aligns with findings from other studies, where peaks lagged between 2 and 6 hours (Sugiyama et al., 2025; Stevens et al., 2022a). Notably, Kamb et al. (1994) observed a similar lag between water pressure and velocity maxima at Columbia Glacier.

## 4.2 Impact of melt-induced speed-up events

Several multi-day speed-up events with a velocity increase of 15-30% above average speed were observed. The period with exceptionally warm temperatures on July 20-21 2024 can be explained by a warm and dry foehn event, characterized by a low relative humidity and a change to north-easterly winds leading to enhanced melt rates and more than a doubling in melt water discharge at the terminus (5c and d). The ice flow velocity reacts rapidly, showing a large speed-up of 28%. Such speed-up events induced by melt water production or lake drainage were documented before. However, events of comparable magnitude on tidewater glaciers are generally connected to lake drainage or heavy rainfall events (Sole et al., 2011; Meier et al., 1994). Melt-induced speed-up events usually showed a smaller velocity increase (Podrasky et al., 2012; Stevens et al., 2022a), with some exceptions (Vieli et al., 2004). Overall, the distinct and sudden response in flow speed highlights the high sensitivity (e.g. rapid and large response) of EKaS to surplus freshwater input in the glacial system.

## 4.3 Impact of a lake drainage event

The peak velocity period around August 10, 2023, occurs during a time of constant, average melt conditions. However, other studies indicated that lake drainages, such as the L2 event, play an important role in forcing the evolution of an efficient subglacial drainage system and often align with the largest speed-up events (Sole et al., 2011). Subglacial discharge from the lake drainage event took about two days to cover the 20 km distance to the terminus. This estimate is based on the timing between the mid-emptying phase observed in Sentinel-1 imagery (Copernicus Data Space Ecosystem, 2025) and the increased plume activity at the terminus. This results in an average discharge speed of approximately 0.1 m/s, which is in good agreement with estimates of down-glacier flood propagation velocities of 0.01-0.1 m/s from previous studies for tidewater glaciers (Vieli et al., 2004; Stevens et al., 2022b; Wehrlé et al., 2024, preprint).

The velocity at EKaS usually decelerates overnight. However, on the night of August 10, 2023, a subdued slow-down over about six hours that even turned into a local acceleration phase in the upper section of the 6 km centreline could be observed (Figs. 2a and 4c). This unusual mid-night behaviour is likely linked to the arrival of the lake discharge water at the terminus area. The potential velocity wave propagation, which is illustrated by acceleration maps in Fig. A7a, is followed by the development of a large plume at the terminus shortly after (Fig. A7b). Given the 6 km distance the discharge wave covered within roughly six hours, a propagation speed of the water wave of about 0.25 m/s can be determined. Our measurements therefore reveal an average propagation speed of 0.1 m/s along the entire 20 km stretch that is increasing to 0.25 m/s at the front. Other studies

**Figure 7.** Satellite-derived, year-round velocity estimates of EKaS' centreline by Gardner, A. S. et al. (2023), as well as TRI time-series of the two field campaigns (green).

confirmed that an efficient drainage system can extent up to several tens of kilometres up-glacier and show a propagation velocity of up to 1 m/s or more (Chandler et al., 2013). After the development of the plume, a fast and distinct decrease in flow velocity occurs over the course of a full day, skipping an entire diurnal acceleration cycle. The velocities fall back to pre-event magnitudes (or even lower). All these observations indicate that the high and rapid discharge of lake water, which started in a distributed flow further upstream, transitioned to an efficient, channelized drainage system while approaching the terminus. Once drainage ceased in this efficient drainage system, subglacial water pressure – and consequently flow velocities – dropped sharply (Das et al., 2008).

#### 4.4 Evolution of basal drainage system



To understand the response of the subglacial hydrology system at EKaS to high-melt or lake drainage events, we need to consider the velocity evolution over the entire year. Fig. 7 shows the intra-annual glacier speed variations, again averaged along the centreline, using satellite-derived data from the NASA MEaSURES ITS\_LIVE project (Gardner, A. S. et al., 2023) for the period between early winter 2023 and end of 2024. Generally, EKaS shows large intra-annual velocity variations with velocities that are more than twice as large in spring than in autumn. More specifically, the speed of EKaS declines during times of high freshwater discharge, leading to a minimum speed by the end of summer. This pattern aligns with a seasonal transition from an inefficient to efficient subglacial drainage system due to increased discharge (e.g. Röthlisberger, 1972; Sundal et al., 2011; Chandler et al., 2013). Thus, EKaS can be assigned to a "type 3" glacier, a classification used by several authors (Moon et al., 2014; Vijay et al., 2021), which is associated with long melt seasons, large meltwater availability and a high intra-annual velocity range. Once the melt period is over, the glacier flow speed at EKaS continuously accelerates, from about 5 m/d to 12 m/d until late spring, when melt sets in again. Our local TRI measurements align well with the annual satellite-derived velocity trends. The diurnal and semi-diurnal velocity fluctuations observed during our two-week field campaigns capture a significant portion of the annual ice flow variability – a level of detail which is not reflected in the sparse and somewhat noisy data provided by (Gardner, A. S. et al., 2023). This is in line with model results by Schoof (2010) showing that short-term water input events can surpass seasonal water pressure signals.






Channelized systems are considered the primary factor controlling the sensitivity of ice velocity to supraglacial water input (Bartholomew et al., 2010). Our observed strong diurnal and multi-day response of the velocity to meltwater input indicates a fast transportation of surface water to the bed (Andrews et al., 2014; How et al., 2017). At EKaS, the correlation between temperature and velocity is lower in 2023 compared to 2024, both for diurnal and multi-day periods. Clearly, overarching processes such as lake drainage events are skewing this correlation. Nonetheless, a major difference between the two years is the timing of field work, which occurred about three weeks earlier in 2024 than in 2023. Thus, the average velocity was still higher in 2024, and the efficiency of the drainage system was likely not yet fully established. This aligns with the current understanding that increased drainage efficiency through the melt season comes with decreasing water pressure at the bed and thus reduced ice speed response to supraglacial meltwater input (e.g. Chandler et al., 2013; Andrews et al., 2014; Doyle et al., 2018). Nevertheless, in both years EKaS remains highly sensitive to short periods of additional water input, even by the end of summer, when an efficient drainage system is already well-developed. This is because the meltwater input is still large enough to surpass over short timescales the capacity of the subglacial drainage system (Bartholomew et al., 2012; Cowton et al., 2013). Together with the findings discussed above, this highlights the glacier's ability to respond rapidly to additional water input, indicating a dynamic basal drainage system.

Most studies on the evolution of basal drainage systems have focused on land-terminating glaciers (e.g. Bartholomew et al., 2010; Cowton et al., 2013; Chandler et al., 2013) or on more upstream regions of tidewater glaciers, where access is easier (e.g. Andrews et al., 2014; Doyle et al., 2018). The bed at the terminus of marine-terminating glaciers, however, is continuously pressurised due to its contact with the ocean, and it is argued that its fast movement might even prevent the evolution of an efficient drainage system, leading to key differences from land-terminating systems (Nienow et al., 2017). Nevertheless, several studies suggest parallels between the two subglacial hydrology settings, namely the evolution of a basal drainage system (e.g. Sole et al., 2011; How et al., 2017), supporting our finding that an efficient drainage system can develop even under persistently high water pressures and fast ice flow throughout the melt season.

To better understand the relationship between air temperature and flow speed, we estimated the temperature-driven subglacial meltwater discharge at the terminus of EKaS (methods in Appendix C), providing indirect insights into the basal drainage system. The comparison between the modelled discharge and the TRI-derived velocity variations (Figs. A8 and A9) indicates that the time-series agree best—showing a clear diurnal pattern—when a water flow velocity of at least 1 m/s is assumed, consistent with values that have been found for an efficient subglacial drainage system (Chandler et al., 2013).

# 4.5 Spatially consistent diurnal velocity fluctuation magnitude

Whereas the flow velocity of EKaS decreases with increasing distance from the terminus, the diurnal or multi-day fluctuations in speed generally exhibit a similar magnitude throughout the entire investigated terminus area between the shear margins (Figs. 4b, 5b, A4, A5). This suggests that short-term velocity variations occur coherently across this area, implying limited short-term internal strain and indicating that the ice responds as a dynamically coupled unit up to at least 6 km upstream. Sugiyama et al. (2025) also found similar speed fluctuation magnitudes along the centreline for melt-water influenced data points, but not for regions close to the terminus, where a high tidal influence was detected. Conversely, several studies observed a clear decay of







velocity fluctuation amplitude in upstream direction from the terminus (e.g. Podrasky et al., 2012; Sole et al., 2011; Stevens et al., 2022a). However, most of these studies – relying on GPS point data along the centreline – focused on locations more than 10 km upstream of the terminus, making direct comparison with our observations difficult (Andersen et al., 2010; Stevens et al., 2022a).

#### 4.6 Sub-daily spatial variability

The sub-daily flow variability exhibits clear spatial inhomogeneity across the terminus area of EKaS. On days with an average or low flow velocity, the transition from acceleration to deceleration (in the evening) or vice versa (in the morning) starts earlier on upstream locations propagating downstream with time (Tab. B1, Figs. 4c and 5c). Conversely, on high-velocity days during multi-day speed-up events, both the acceleration and deceleration generally starts earlier at the terminus compared to locations further upstream. Additionally, these upstream propagating velocity changes often occur in a distinct block-wise spatial pattern, which align with major crevasses or rifts of the glacier.

To further analyse the switch between the two transition regimes, a set of equally spaced points was selected along the centreline and corresponding velocity and acceleration 2024 time-series are illustrated in Fig. 8. Again, in the upstream transition regime, points at the terminus change from positive to negative acceleration (e.g. crossing the zero-line of Fig. 8b) earlier than locations further upstream; in the downstream transition regime, upstream locations undergo this acceleration sign change before the terminus does. Additionally, Fig. 8b shows that during days with a downstream propagation, the glacier uniformly accelerates/decelerates throughout the entire centreline. In other words, while the timing of the transition may vary slightly, the magnitude of acceleration remains almost constant along the entire centreline, with values usually within  $\pm 2 \,\mathrm{md}^{-2}$ . In contrast, on high-velocity days, when the transitions propagate upstream, the acceleration/deceleration magnitude is generally much higher at the terminus, with values largely exceeding  $\pm 5 \,\mathrm{md}^{-2}$ , than further upstream, where values are generally within  $\pm 3 \,\mathrm{md}^{-2}$ . To conclude, the most significant difference between the two transition regimes is the glacier's behaviour at the terminus, where acceleration/deceleration values on high-velocity days are much higher compared to locations further upstream.

On days with a low or average flow velocity, when a downstream propagating transition is typically observed, the diurnal increase in melt water input at EKaS seems to be sufficient to surpass the capacity of the subglacial hydrological network, resulting in a pressure peak at the bed and a subsequent speed-up of the glacier (Bartholomew et al., 2012; Cowton et al., 2013). Model results showed that such short-term spikes in water input still manage to increase the water pressure leading to periods of fast ice flow, even in a channelized drainage system (Schoof, 2010). Assuming a similar melt input throughout the terminus area of EKaS, a less efficient drainage system further upstream would become pressurized first after the onset of melting, leading to a downstream propagating pressure wave as the meltwater flows down towards the terminus. This would cause the upstream area to accelerate earlier than the downstream parts, as observed on the majority of the days on EKaS (Tab. B1). Accordingly, once the melt water input decreases in the evening, the discharge reduces first in the large upstream areas, leading to a drop in water pressure (Meier et al., 1994). This would results in a downstream propagating deceleration wave and explain the observed delayed transition from acceleration to deceleration near the terminus compared to the upstream region.

Figure 8. a) LOS velocity and b) its acceleration time-series at different positions along the 6 km long centreline in 2024.

The contrasting behaviour of upstream propagation during speed-up events, including the block-wise patterns, seems less straightforward to interpret. We expect that the much higher water input to the basal drainage system on these days strongly overwhelms the capacity of the subglacial network. Consequently, basal water pressure is assumed to sharply increase towards approaching flotation. This is particularly pronounced near the front, where the bed is already under high pressure from contact with the fjord water and may, for a brief period, lead to basal sheet flow due to basal separation and the hydraulic jacking mechanism (Röthlisberger and Iken, 1981; Cowton et al., 2016; Sugiyama et al., 2025). This is also indicated by the much stronger velocity increase near the terminus than upstream and the patchy acceleration pattern (Fig. 8b). The observed behaviour seems to be in line with the model results by Pimentel et al. (2017), showing that the rapid injection of water, after reaching a certain threshold drainage, completely overthrows existing drainage systems up to a certain distance from the terminus. As soon as the water input decreases, the basal water pressure seems to collapse, again in this block-wise pattern, leading to abruptly decelerating basal sliding at the terminus, which then propagates upstream. On these days, the described process fully overwrites the downward propagating diurnal signal discussed above, and precludes a uniform acceleration and deceleration reaction of the glacier as observed on low-velocity days.

## 5 Conclusions

Terrestrial radar interferometric observations revealed high spatio-temporal ice velocity variations in the lowest 6 km of Eqalorutsit Kangilliit Sermiat's terminus during two separate two-week periods in the summers of 2023 and 2024. We found




that the glacier's velocity shows clear temporal fluctuations on both diurnal and multi-day scales, primarily driven by additional water input from surface melt or lake drainage events. Tidal changes and ice mélange cover show no detectable influence on the velocity variations at diurnal time scales. The diurnal velocity peaks about four hours after the temperature and exhibits a diurnal peak-to-peak amplitude of about 7% of the mean speed. Thus, the melt water of each day seems to quickly access the basal system, which leads to a peak in basal water pressure and thereby a temporally enhanced sliding velocity. During a high-temperature period over multiple days caused by a foehn event, the ice speed was observed to increase by 28%. Additionally, several detected multi-day events with substantially enhanced flow velocities could clearly be linked to subglacial or marginal lake drainage events. These findings underscore the glacier's sensitive velocity response to additional water input and its relatively rapid adjustment, indicating an efficient and dynamic basal drainage system with a persistently high basal water pressure.

While the glacier shows increasing velocities towards the front, the diurnal fluctuation amplitudes remain remarkably consistent along the 6 km long terminus area. However, when considering the spatial evolution of the diurnal velocity variations, local differences in the propagation pattern become apparent. On days exhibiting average diurnal velocity fluctuations, the glacier starts to accelerate and decelerate in the upstream regions first, with the signal subsequently propagating down towards the terminus. Conversely, on days characterized by high flow velocities (during multi-day speed-up events), the acceleration and deceleration was observed to start at the terminus propagating upstream over time. Interestingly, in the lowest few kilometres, these upstream propagating velocity changes occur in a distinct block-wise spatial pattern (bordered by rifts), indicating patch-wise adjustment in basal conditions. Our observations suggest that rapid meltwater input temporarily overwhelms the basal drainage system near the terminus, causing local sheet flow and strong flow acceleration, followed by a rapid, block-like collapse in basal water pressure and sliding as water inputs subside.

We conclude that the studied tidewater glacier, despite having established an efficient drainage system towards the end of the melt season, remains very sensitive and reacts fast to surplus water entering its basal system. Thus, our high spatio-temporal velocity observations highlight the profound influence of water input and basal hydrology on the short-term, small-scale flow dynamics of tidewater glaciers near the terminus.

*Code and data availability.* The code and data used to produce the figures in this manuscript can be accessed here: https://github.com/ArminDach/TRI\_velocity\_EKaS.git. Raw TRI data are available upon request.

**Figure A1.** Comparison of the air temperature record 2023 used in Fig. 2 from near the fjord east of the terminus of EKaS to the Mittafik Airport weather station in Narsarsuaq operated by the Danish Meteorological Institute (Drost Jensen, 2023).

**Figure A2.** Comparison of the air temperature record 2024 used in Fig. 3 from near the hill on the opposite side of the glacier terminus to the AWS at the fjord east of the terminus of EKaS and to the Mittafik Airport weather station in Narsarsuaq operated by the Danish Meteorological Institute (Drost Jensen, 2023). Note that parts of the fjord temperature signal is suppressed by local inversion effects.

# Appendix A: Figures

**Figure A3.** Sentinel-2 imagery before and after the marginal lake drainage event L2 (red) occurring between August 7-9, 2023, about 20 km upstream of the terminus. The Sentinel-2 inset map also shows the location of the subglacial lake L1 (pink) from the western tributary (Copernicus Data Space Ecosystem, 2025).

**Figure A4.** a) Average LOS velocity time-series in 2024 for three parallel flow lines and b) their deviation from the central flow line highlighting the spatially constant diurnal velocity fluctuations. The image in the inset graph is a Sentinel-2 acquisition from Aug 7, 2023 (Copernicus Data Space Ecosystem, 2025).

**Figure A5.** a) Average LOS velocity time-series in 2024 for three parallel transverse lines and b) their deviation from the southern transverse line, highlighting the spatially constant diurnal velocity fluctuations. The image in the inset graph is a Sentinel-2 acquisition from Aug 7, 2023 (Copernicus Data Space Ecosystem, 2025).

**Figure A6.** LOS shear strain rate during the evening of July 19, 2024, highlighting the large lateral velocity gradient as well as the rift positions that potentially influence the block-wise acceleration and deceleration pattern.

**Figure A7.** Potential lake drainage event L2 approaching the terminus on August 10, 2023 a) leading to a weakened mid-night deceleration (lower part) or even an untypical acceleration (upper part), b) followed by the development of a large plume visible at the terminus in the time-lapse imagery from 6:00 am onwards (green rectangle). The number in the box shows the local Greenlandic time. The background image in a) is a Sentinel-2 acquisition from Aug 7, 2023 (Copernicus Data Space Ecosystem, 2025).

**Figure A8.** a) Average LOS velocity time-series in 2023. b) Air temperature-based modelled total meltwater discharge at terminus of EKaS in 2023 for surface meltwater velocity of 0.1 m/s, 1 m/s and 2 m/s.

**Figure A9.** a) Average LOS velocity time-series in 2024. b) Air temperature-based modelled total meltwater discharge at terminus of EKaS in 2024 for surface meltwater velocity of 0.1 m/s, 1 m/s and 2 m/s.

**Table B1.** Categorical overview on the spatial propagation pattern of the diurnal acceleration in the morning and deceleration in the evening during the two fieldwork periods in 2023 and 2024.

| 2023   |                   |                   |
|--------|-------------------|-------------------|
| Date   | Morning           | Evening           |
| 3 Aug  | no data           | no data           |
| 4 Aug  | downstream        | unclear           |
| 5 Aug  | downstream        | no data           |
| 6 Aug  | no data           | no data           |
| 7 Aug  | no data           | upstream          |
| 8 Aug  | upstream          | unclear           |
| 9 Aug  | unclear           | downstream        |
| 10 Aug | deceleration only | deceleration only |
| 11 Aug | downstream        | downstream        |
| 12 Aug | downstream        | downstream        |
| 13 Aug | unclear           | downstream        |
| 14 Aug | downstream        | upstream          |
| 15 Aug | unclear           | no data           |
|        |                   |                   |

| 2024   |            |            |
|--------|------------|------------|
| Date   | Morning    | Evening    |
| 12 Jul | unclear    | upstream   |
| 13 Jul | unclear    | downstream |
| 14 Jul | downstream | downstream |
| 15 Jul | upstream   | downstream |
| 16 Jul | unclear    | downstream |
| 17 Jul | unclear    | downstream |
| 18 Jul | downstream | downstream |
| 19 Jul | downstream | upstream   |
| 20 Jul | upstream   | upstream   |
| 21 Jul | unclear    | unclear    |
| 22 Jul | unclear    | downstream |
| 23 Jul | downstream | unclear    |
| 24 Jul | downstream | unclear    |
| 25 Jul | unclear    | no data    |

Appendix B: Table

# Appendix C: Modelled meltwater discharge

To model meltwater discharge at the terminus, a Temperature Index Model (Braithwaite, 1995) was applied to calculate melt at each pixel. Then, the melt was added up considering the flow distance to the terminus. In more detail, hourly air temperature data from the Mittafik Airport weather station in Narsarsuaq (Drost Jensen, 2023) were used as input after correction for elevation using the average lapse rates for summer (-0.0051 °Cm<sup>-1</sup>) and winter (-0.0092 °Cm<sup>-1</sup>), which is derived from the on-ice Q-transect meteorological stations QAS\_L, QAS\_M, QAS\_U (How et al., 2025).

The continuous ablation measurements from the DWIAT sensor of Forloh (2025) located on the Nordbo outflow lobe of EKaS 25 km upstream of the calving front was used to calibrate the ice melt resulting in the following melt factor: mf\_ice = 0.00017 mh^{-1}°C^{-1}. The snow melt factor mf\_snow = 0.0001 mh^{-1}°C^{-1} was obtained by calibration using the freshwater discharge product by Mankoff et al. (2020b). Snow line elevations and hence snow extent were extracted from Sentinel-2 imagery (Copernicus Data Space Ecosystem, 2025), interpolated between dates, and used to distinguish between snow or ice melt for each pixel and timestep.

The flow distance was derived using a flow routing routine based on basal hydraulic potential, estimated from bed topography and ice thickness data (Morlighem et al., 2017) under the assumption of water pressure at ice overburden pressure (Shreve, 1972). The TopoToolbox (Matlab, 2014) was applied to compute flow accumulation, catchment area, and the flow distance of each pixel to the glacier front. The received catchment aligns with that of Mankoff et al. (2020b).

Discharge velocity over ice was estimated using Manning's equation (Arnold et al., 1998) and the mean surface slope from the 50 m resolution ArcticDEM (Porter et al., 2023), yielding an average surface meltwater velocity of 2.4 ms<sup>-1</sup>. As this value neglects refreezing and subsurface flow, likely overestimating actual conditions, we used velocities of 2 ms<sup>-1</sup>, 1 ms<sup>-1</sup>, and 0.1 ms<sup>-1</sup> (Figs. A8 and A9), acknowledging that actual values likely vary spatially and seasonally.

Author contributions. AD, AKW and AV conceived and designed the study. AD, AKW, DG and AV conducted the field work. AKW and AD
 managed the data processing. AD performed the data analysis, drafted the manuscript, and interpreted the results. All authors participated in the result discussion and writing process of the final manuscript. All authors have read and approved the final version of the paper for publication.

Competing interests. The authors declare that they have no conflict of interest.

Acknowledgements. We are grateful to the Greenlandic community for welcoming us to their beautiful country. We thank Sebastian Rosier,

Antonin Salamin, Diego Wasser and Ethan Welty for all their support during the field campaigns. The authors are grateful to further members
of the field campaign, in particular Manuela Köpfli, Brad Lipovsky, Enrico van der Loo and Selina Wetter. Additionally, we thank Martin
Lüthi and Adrien Wehrlé for their support and fruitful discussions during the data analysis process. We acknowledge Forloh (2025) for

making continuous ablation data available. This study is part of the SPI flagship initiative GreenFjord funded by the Swiss Polar Institute (project number SPI-FLAG-2021-002), which also provided valuable support with field safety and instrument permits.

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
