# Peer review of "High spatio-temporal velocity variations driven by water input at a Greenlandic tidewater glacier"

_EGUsphere, 2025_

## Referee Comment (RC1)

**Review of egusphere-2025-5193:**
**High spatio-temporal velocity variations driven by water input at a Greenlandic tidewater glacier**

William David Harcourt, University of Aberdeen

December 19, 2025

**Summary**

The study of Dachauer et al. (2025) utilises a high temporal resolution terrestrial radar data set of a tidewater glacier in south Greenland to study its response to meltwater inputs to the glacier system and the impact on ice flow. They show that, even at the end of the summer season when the subglacial hydrological system is likely predominantly efficient, that diurnal fluctuations in ice velocity correlate to changes in air temperature, suggesting that the system remains active and sensitive to meltwater inputs. Furthermore, during periods of reduced ice velocities, the glacier sped up in response to the drainage of a lake (which I assume was an ice-marginal lake although the text was a bit unclear on this). Finally, the authors suggest that the diunal signal propagates from the upper glacier to the terminus on slow flow days, but during velocity speed up events the diurnal signal propagates from the terminus and up glacier. The results demonstrate the sensitivity of the glacier to meltwater inputs over shorter timescales even in late summer when the subglacial system has transitioned to a predominantly efficient system.

**General Comments**

I found this paper to be an interesting read and the analysis that has been done nicely highlights the interesting dynamics associated with the glacier. Measurements such as these are hard-won - it takes significant logistical planning and often long periods in the field collecting the measurements. The authors should be congratulated on generating such a useful data set! I have provided technical comments below that ask for clarification in several areas. I have a few more substantative comments that I ask the authors to consider in their revision:

- Methods: The methods are insufficiently described in the paper. Although the TRI measurements have been widely described in previous studies, the paper still needs to include a description of the specific set up used in this study, including key equations to derive displacement measurements, derive 3D data (i.e. DEMs) and other processing steps e.g. coregistration (e.g. due to wind buffeting of the radar).

- The upward and downward propagation of the diurnal velocity signal is not clear to me. After looking at Figs. 4 and 5 several times I can start to see it, but I think it needs a bit of explanation. You could use an example of where the upward/downward propagation signal is clear in a single figure - you have attempted this in Fig. 6, but it would be useful to also show the 1D profiles as an example. My broader point is that these are referenced several times in the discussion and it

is sometimes very hard to see it - case in point is Fig. 8 where these propagation patterns are mentioned several times but I struggle to see the patterns.

- The implications of the study are not effectively described in the context of Greenland Ice Sheet dynamics and future changes. Part of the problem here is that a research question is not stated in the introduction, so it currently reads a bit more as a description of results and interpretation of them. These then need to be combined to provide an holistic overview of the glcier dynamics. I.e. the overarching conclusion is that the glacier is sensitive to meltwater inputs in late summer when the subglacial system should have an efficient drainage system. Quantifying the sensitivity (see comments below) would enable you to compare this to other outlet glaciers, but also whether the processes observed at EKaS are representative of other regions. Articulating a clear process-driven framework for the processes and then expanding that view to the whole of Greenland would be a nice way to summarise the data in a way that brings all the analyses and interpretations together.

**Technical Corrections (References to line (L) numbers in preprint)**

L1-2: Aren't ice discharge and frontal ablation essentially considered the same?

L5: If the data were gridded on a uniform grid, you should state the resolution precisely.

L6-7: Sensitivty in terms rapid change in velocity in response to meltwater inputs? Or something else? Be precise.

L13-14: You may touch on this later, but it would be useful to state a clear outcome of the paper succinctly - the relationship between meltwater and dynamics is complex and widely studied, so what exactly does this study find that contributes to this knowledge?

L16-18: Glaciers are still the largest contributor, check out the recent GLAMBIE community estimate (https://www.nature.com/articles/s41586-024-08545-z). There are also more recent references that should be cited alongside these e.g. Otosaka et al. (2024)

L20: I wouldn't say it is limited? It's a very active research area! I would instead tease out the key processes of most interest e.g. meltwater and basal sliding feedbacks, ocean thermal forcing, precipitation changes.

L23: E.g. studies such as Tedstone et al. (2015) [https://www.nature.com/articles/nature15722] that showed long-term slowing down of a land-terminating glacier in response to larger melt input to the bed.

L28: Optical and SAR are imagery, I think you mean feature speckle tracking in optical and SAR image pairs

L30: Far more than a few studies.

L32-34: Remove the tidal forcing part from this sentence - use a separate sentence to discuss tidal impacts on velocityies.

L34-35: Broad statement that is not always true. I think its important to acknowledge the complexities here e.g. drainage may shit down over winter (not always if meltwater can be stored at the bed). in spring, there may be a spike in velocity due to a sudden input of meltwater, then at the end of summer velocities may decline. Diurmal changes can be observed, whilst long-term changes are less well known due to observational constraints.

L36: Challenges? I assume you are referring to measuring the inaccessible bed?

L39-40: Sentence needs to be qualified with the reasons why we need short-term and high resolution observations. To observe diurnal patterns? Response to calving?

L58-63: It would be good to highlight other GPRI / terrestrial radar studies to investigate tidewater glacier dynamics as there are now quite a few. You could dedicate a whole paragraph to this, or at least a few sentences, particularly discussing the benefits and limitations of the approaches and processing complications.

L60: What is the research question being addressed?

L74: Could you put this into perspective with other glaciers? How does it compare to the big tidewater glaciers in Greenland e.g. Jakobshavn, Helhim etc.?

L84: Do you have a picture of the set up? Could add this to Figure 1. It's to confirm visually the field of view.

L87-89: This should go in the data processing section, including a slightly expanded explanation of how thr DEMs were generated and then the DEM processing.

L99: Did they capture the regional temperature trends? Figs A1 and A2 seem to, but would be good to qualify this sentence with a statement saying that they do and any biases.

L118: No mention of how the DEMs were derived?

L119-127: Although the method is well described in previous studies, I think you do need to discuss a bit more detail about the interferometric approach inc. key equations of how you related phase changes to displacement. Did you have to coregister the images e.g. due to winf buffeting of the radar?

L120-121: Reword: 'The TRI transmits from a single antenna and measures radar backscatter using two receiver antennas.'

L124: Important recognise throughout the paper that the temporal resolution of your measurements is 30 minutes and NOT 1 minute, as is claimed.

L147: Although this section is generally well-written, it was sometimes unclear when you were discussing 2023 and 2024 data. E.g. the peak-to-peak amplitudes were stated to be 0.5 m/d, but what year? I would discuss each year in turn for clarity (within each section though, I like having the results section split into themes as you have done).

L168: I note here that the method to extract mélange presence / absence has not been described in the methods, but should be. Some for the detection of plumes.

L174: It's slightly off that these are not disucssed in chronological order i.e. 2023 and then 2024.

L185: State dates.

L190: Reference relevant section / figures.

L194: Isn't it an ice-marginal lake? Reference the figure where it is labelled L1 - Figure A3 I think.

L195: How was this calculated?

L196: Is there a figure showing the lake drainage event?

L198: When did the lake drain? Do you have exact dates?

L203: Reference relevant figure. Wasn't L2 a supraglacial lake?

L204: True if it's a supraglacial lake, unclear if it's an ice-marginal lake.

L212-222: As noted above, a description of the methods used to calculate these should be provided. Are volumes calculated for all calving events? Might be useful to add these to Figures 2 and 3.

L218-222: What about calving and diurnal variations - are there any signals or correlations?

L230: I assume the contant difference means the dirunal signal is clear at all distances from the terminus? I'd explicitly state this.

L234-242: The primary point here is that diurnal fluctuations propagate from the top of the centreline profile to the terminus on low velocity days, whereas on high velocity days it propagates from the terminus. However, in 2023 this is not clear as the green boxes appear before the high velocity days. In 2024, there does seem to be a correlation. So, based on the data available, this assertaion may be partially true, but I think its worth stating that this may only hold when the 'high velocity days' are significantly larger than the base velocity level.

L239: Were the orange/green boxes calculated manually? Or some other way?

L243-252: Can you label to the side of the maps which represent downward propagating and upward propagating diurnal velocity variations. This section is a bit hard to read because you reference velocity speed up events (which relate back to Figs 2 and 3) but you showing acceleration maps, so it is not clear where I should be looking. You also mention the 'outlines' of the block like pattern of flow, but this should be labelled for clairty - I assume it relates to the step-like pattern clearly visible in Fig, 6a.

L255-261: I would start with a comparison to other Greenlandic outlet glaciers and then consider other glaciated regions (e.g. Alaska).

L273-275: Does the July 23 precipitation have any relationship to the velocity increase on July 25th?

L276-282: This is quite vague - how would wing speed affect the velocity variations? For cloud cover, I suspect it is not a straight 1-1 relationship, does you RH data provide any clues as to the relationship?

L302-304: I must admit, I am struggling to see the acceleration. Is it the multi-day slowdown that is important here?

L310-315: If the water is being channelised and propagating along the bed, would there be spatial velocity variations - i.e. high flow over the channelised region, slow flow over the less non-channelised sections (assuming no water here, or maybe there will be some form of distributed drainage system?). I guess I am looking to see here a bit more underpinning theory related to the observation rather than simply stating water flow down and then the ice flow increased.

L322: To prove this point, you need a graph over the same period showing frshwater discharge - I assume you more broadly mean 'melt' i.e. a temperature graph would suffice?

L326-327: You might discuss thebelow, but this part is interesting - stored meltwater at the bed over winter? Possibly, considering the recent findings of Hansen et al. (2025).

L344-245: Is it possible to calculate average diurnal melt rates and therefore volumes that are transferred to bed? This might allow you to quantify the sensitivity of the bed to extra meltwater inputs by comparing with the velocity changes over the same time period - I would expect to see a declining sensitivity over summer. Ah, I see you have done in Appendic C! But can you use this to quantify the sensitivity of the drainage system over the velocity time series in Figure 7?

L376-388: It is difficult to relate the text to areas on the figure, probably because the changes are subtle and/or embedded within the multiple lines shown in Fig. 8b. Can you help to identify these locations more clearly e.g. lines on the graph, annotations? This would certainly help improve the clarity of reading in this section.

L377: 'were selected'

L378: 'time-series extracted and shown in Fig. 8'

L377-378: Maybe I missed it, but I am not sure thge term 'upstream transition regime' has been explicitly stated as this before.

L381-383: Not clear to me what this relates to in Fig. 8b?

Figures

Figure 4: Might be more intuitive to have the brighter colours representing higher velocities.

Figure 8: Can you add a colorscale indicating the distance from the terminus?

---

## Referee Comment (RC2)

**Review of egusphere-2025-5193: High spatio-temporal velocity variations driven by water input at a Greenlandic tidewater glacier**

**Summary**

This manuscript investigates short-term ice-flow speed variability at Eqalorutsit Kangilliit Sermiat, South Greenland, over sub-diurnal to multi-day timescales using Terrestrial Radar Interferometer (TRI) measurements. The authors document clear diurnal and multi-day velocity variations that are associated with episodic water inputs, including surface melt, supraglacial lake drainage, and subglacial or marginal lake drainage events. A significant difference between the two transition regimes is the glacier's response at the terminus, where acceleration and deceleration during high-velocity periods are larger than at locations further upstream. When water input decreases, basal water pressure appears to collapse in a block-wise pattern, resulting in abrupt deceleration of basal sliding at the terminus that subsequently propagates upstream. During such events, this mechanism overrides the downward-propagating diurnal signal observed during low-velocity periods and prevents the more uniform acceleration and deceleration behavior of the glacier.

The study demonstrates the strong potential of TRI observations to resolve short-term velocity changes, and the resulting dataset is highly valuable. Overall, the interpretations are well aligned with the scope of The Cryosphere and provide meaningful insights into short-term glacier dynamics. I have some suggestions to improve this manuscript, as below.

**General Comments**

- The advantages of using TRI are not sufficiently articulated. In particular, the rationale for the chosen temporal resolution (30 minutes) and cut-off period (3 hours) is unclear. Please provide justification for these choices, including discussion of coherence, signal-to-noise considerations, and any thresholds applied.
- L58-63: This is not the first study to use TRI for this type of analysis. Additional relevant literature should be cited (e.g., Drews et al., 2021; Holland et al., 2016; Voytenko et al., 2015; Xie et al., 2018, 2019).
- L77-86: The TRI azimuth resolution varies with range. If the region of interest is within 6 km, what is the effective resolution near the ice front and at upstream locations? While the resolution of 6.9 m at 1 km and the maximum system range of 16 km are stated as system specifications, more detailed information is needed for the actual region of interest. Please also specify the multilook factors applied and explain why these choices were made.
- L104: The plume extents are classified as (1) none, (2) small, (3) medium, and (4) large. Please clarify the criteria or thresholds used for this classification. Was this done

qualitatively (e.g., visual inspection) or quantitatively? A similar clarification is needed for L168 regarding the definitions of small, medium, and large mélange-covered areas.

- L236: The statement describing a lag of approximately two hours between the terminus and the location 6 km upstream is unclear. For example, deceleration appears to occur earlier and more strongly at the terminus on 2023-08-10, 2024-07-15, and 2024-07-20. Please clarify how the lag was determined and how it varies among events.

- L243-252: The description of Figure 6 would benefit from clearer visual guidance. Please consider annotating the figure (e.g., with arrows or highlighted regions) to indicate the areas discussed in the text, as the current presentation is not very intuitive.

- L262: The manuscript states that a Fourier analysis of the two velocity time series revealed strong 24 h peaks for both years, but the results are not shown. To support the subsequent conclusion that this reflects a solar rather than lunar influence (12.4 h), the Fourier analysis should be presented, at least in the Appendix.

- L274: The manuscript suggests that the relatively low precipitation rate resulted in only a small velocity response. Please clarify the basis for this interpretation. Is this because precipitation does not immediately translate into enhanced basal lubrication? Is there any relationship between precipitation and the acceleration observed on 2024-07-24?

- L298: If Sentinel-1 imagery is not shown or analyzed in this paper, please clarify why it is mentioned here, or include the corresponding results in the Appendix.

- L303: The manuscript refers to acceleration during the night of 2023-08-10. However, this pattern is not clearly shown in Figures 2a and 4c. Please clarify this interpretation or revise the text and/or figures accordingly.

- L369: Several previous studies have used GPRI for similar analyses, rather than relying solely on GPS point measurements along the glacier centerline. These studies should be cited, including Kane et al. (2020), Xie et al. (2018, 2019), and Drews et al. (2021).

**Figures**

In general, the figures, axis labels, tick labels, colorbars, and legend texts are too small and difficult to read. Please enlarge them to improve clarity.

- Figure 7: This figure appears to add limited new information to the manuscript. Consider moving it to the appendix.

- Figure A3: Please consider also showing the Sentinel-2 images corresponding to drainage event L1.

- Figure A4 and A5: Including 2D spatial velocity maps as animations or videos would be more effective in illustrating diurnal velocity fluctuations.

- Figure A7: Please consider overlaying the glacier centerline to improve interpretability, particularly in panel (a).

**Minor comments**

- The manuscript uses both "ocean-terminating glacier" and "marine-terminating glacier." Please choose one term consistently throughout the text (or, "tidewater glacier").
- L69: The statement that the glacier's terminus has advanced over the past few decades should be quantified (e.g., by how much).
- L73: Citation error (also occurs at L330).
- L125: The statement "which has a sensitivity of less than 1 mm" requires a reference.
- L206: Please consider showing the Sentinel-1 data, at least in the Appendix.
- L237: "Fig. 4c and Fig. 4c" typo.
- L287: "Fig. 5d" no such panel exists.

---

## Community Comment (CC1)

**Community comment to Dachauer et al., TCD, "High spatio-temporal velocity variations driven by water input at a Greenlandic tidewater glacier"**

Dear Armin Dachauer and co-authors,

My name is Christian Wild, and I am a geophysicist at the University of Innsbruck, Austria. I have experience working with terrestrial radar interferometry (TRI) in both polar and alpine environments and have recently discussed *Dachauer et al.* in my MSc-level literature seminar. Our group's discussion raised a number of stimulating points that we believe could contribute to further strengthening the presented conclusions. We outline our comments below and hope that our feedback will be useful to the authors.

**Summary:**

The study investigates short-term velocity variations (sub-diurnal to multi-day) at the terminus of the tidewater glacier Eqalorutsit Kangilliit Sermiat in South Greenland. Using a terrestrial radar interferometer (TRI) with high spatial (metres) and very high temporal (1 minute) resolution, the authors capture line-of-sight velocity changes and relate them to hydrological forcing. Sub-diurnal velocity variability is associated with surface melt, and multi-day speed-up events are linked to lake drainage. Important conclusions are made towards the mechanistic link between structural controls (rift systems) modulating hydrologically induced flow.

This dataset is exceptionally rare, and we acknowledge the considerable logistical and environmental challenges required to collect it. At the same time, the richness of the observations implies that additional analyses could help test the robustness of the proposed mechanisms and extract even more insight. Our comments are offered with the intention of helping the authors make the most of the already-processed data and the code they have available, without the need to constrain an ice-dynamical model with their observations to support some of their conclusions.

**Strengths:**

- The novel resolution provided by the TRI enables unprecedented detail of glacier dynamics, which are outside the resolution of conventional satellite or UAV campaigns, Its richness offers substantial potential for further scientific insight.
- The study reveals that the glacier's response to hydrological forcing is more complex than a simple uniform sliding mechanism, with contrasting behaviour observed up-glacier versus down-glacier. The possibility that existing rift networks modulate these responses is both important and intriguing, and represents a meaningful scientific contribution.

• TRI data processing is inherently complex and often difficult to convey in a clear and accessible manner. The authors demonstrate that they mastered the method, and present the workflow in a way that is both understandable and reproducible. The figures are clear and thoughtfully constructed. In particular, the creative Figure 8 is especially effective in illustrating and synthesizing the key ideas discussed, and deserves special acknowledgement for how well it supports the interpretation.

**Weaknesses:**

- In its present form, the paper reads as observational rather than improving our understanding on a system-level or the broader context. The paper could connect subglacial hydrology more directly to implications for mass loss, calving behaviour and interactions with ice melange, plumes versus terminal velocity (Helheim Glacier, Melton et al., 2022) or even sea-level projections. One possible approach, with the processed data at hand, is to look along ice-cliff transects (or even better delineating the rift clusters, a.k.a. blocks for more detailed analysis of the proposed mechanisms). There is a figure buried in the SI, to investigate the proposed mechanistic link without the need for modeling experiments. Figures 2 and 3 are getting there, but are then disregarded in the text in this context.
- The TRI data set is not fully explored. The advantage of the system is its high spatio-temporal resolution (as reflected in the title), but temporal downsampling from the TRI's native resolution of 1 min to 3 h, and lack of spatial heterogeneity of the presented analysis (flow-parallel transects, versus possible 2d fields) raises doubt about the robustness of the results. Further analysis of shorter/longer temporal baselines, especially the evolution of rocky areas and coherence, would greatly strengthen the conclusions about the subglacial hydrology, by excluding atmospheric variability, or spatially heterogeneous ablation, as alternative mechanisms captured by the system.

**Major comments:**

1. Why are rocky areas excluded from the analysis? The TRI illuminates not only the moving ice surface, but also relatively stagnant rock faces and slopes. These areas can be used to assess the level of background noise, but even more importantly also to clearly distinguish between real glacier acceleration and apparent motion due to atmospheric variability. If the detected accelerations of the ice surface (up/down glacier and vice

versa) also occur on the rocky areas nearby, clearly they are driven by atmospheric variability, if not, the author's reasoning in Section 3.2.2 are strengthened. Mean velocity fields (Fig. 1) and mean centerline velocity averages (Figs. 2 and 3, averaged over the first 2 km, but excluding a 100m section right at the terminus. The justification for using only this part remains rather vague. See Minor comment 1) are presented, but not their standard deviations. How does LOS velocity variability change over time and space? Is there a connection to range when looking at the mean standard deviation of each pixel? How does velocity variability compare to the determined velocity increase of 15-30% above average speed?

- 2. Temporal downsampling. How robust are the results when shorter or longer temporal baselines are used to calculate interferograms? How was the presented temporal baseline of 30 min chosen in the first place? We understand that TRI data were acquired at 1 min intervals, stacked to 30 min and then smoothed using a 3h low-pass filter to derive the presented values, which are then correlated with the time series of air temperature and relative humidity. Wouldn't it make more sense to similarly smooth the AWS data for a better comparison to the TRI-derived numbers? Moreover, the propagation speed of radar waves is controlled by the absolute amount of water vapour in the air (Goldstein 1995) so using relative humidity is not an ideal quantity in this regard, we suggest to use absolute humidity or specific humidity, as typically used to study adiabatic processes in atmospheric science. Other processing choices of the TRI were assumed to be well established, but they are not yet. How were these determined? In particular, can you provide reasoning for your choice in the temporal baseline, stacking interval for averaging, filtering, and geocoding.
- 3. Some major conclusions read deterministic (e.g., L416 'driven by additional water input from surface melt', or L422 'could clearly be linked to lake drainage events') without direct observations in the subglacial system, such as ground-penetrating radar or additional evidence from modeling experiments. Does the temporal coverage of the TRI data set justify these conclusions, if only the end of a drainage event is captured? In contrast, these core conclusions are only made from 'weak' temporal correlation (L158: 0.6 at a lag of 4 hours) without rigorous analysis of their statistical significance. We suggest moving these conclusions, which are only partly warranted by the data analysis, to a dedicated discussion section, if no further evidence from auxiliary analysis are presented.
- 4. Given the spatial detail resolved by the TRI measurements, we would appreciate a more in-depth examination of how the rift system mechanically influences hydrologically driven flow near the terminus. Previous studies (e.g., Ultee et al., 2022) generally assume

spatial homogeneity when assessing terminus variability, but here we have the opportunity to investigate this signal at much finer spatio-temporal scales. We therefore recommend focusing the analysis on ROIs where individual blocks exhibit distinct behavior, and we request an additional figure showing the locations of these blocks and their respective LOS velocity evolution over time.

**Minor comments:**

- 1. TRI data processing: How was the geocoding performed? What DEM? Temporal baseline, how was this determined to be a representative time scale? What pixel was used for the phase unwrapping, and how was it chosen (coherence?)? What is the temporal variability of the phase at the chosen pixel and how does that add to the overall uncertainty? L134: 'Where data quality is highest' reads subjective, is there a threshold in coherence applied? L82 please add or reword to mentioning range/azimuthal resolution at the actual study site, which is 3-9 km away.
- 2. General level of processing: How was the plume size determined? Was there a quantitative component to the use of "small", "medium" or "large"? L87: Were calving events detected manually, semi-automatically or automatically from the TRI derived DEMs? Similar for time-lapse imagery (L104), how was ice melange extent and subglacial plume extent quantified? L106: Add when the subglacial lake drainage event occurred approximately to help the reader.
- 3. L114: Interpretation of tides, how well is the measured tide captured by a tide model such as Greenland 1km Tide Model (Gr1kmTM, Howard and Padman, 2021)?
- 4. L158: Are the following correlations statistically significant? How about any orrelations to absolute/specific humidity?
- 5. L178: Could this be a low-pass filtered humidity signal? A comparison to a nearby rocky area, and its apparent velocity variability, might further support the observation of a real response of the ice to melt triggered by the drastically high air temperatures.

- 6. L225: Why is the analysis constrained to a centerline, and not the full spatial resolution of the TRI utilized and explored?
- 7. L226: 'typical' always needs at least one e.g. type citation. Similar to the word 'generally' elsewhere.
- 8. L237: 'Fig. 4C' typo.
- 9. L238: How do rocky areas behave during these times? Are they showing similar oscillations as observed on the glacier?
- 10. L255: Quantify 7% of mean velocity and compare it to the uncertainty of the TRI measurement.
- 11. L287 why cite Fig. 5c and d for foehn events? Double check the figure references. There is no panel 5d.
- 12. L327: Agree that the mean velocity fits well, but the TRI variability is much larger than the satellite-derived variability at this time of the year. Given that the TRI measures only the LOS component, one would expect a smaller mean and smaller variability when compared to the satellite? Rotating the satellite derived velocity into LOS would be beneficial to this comparison, which requires geocoding of the TRI.
- 13. L371: The sub-daily pattern could also be atmospheric variability on a daily cycle, couldn't it? Again, a comparison to nearby rocky areas would help to strengthen the observed response of the glacier.

**Figures**

• General: Label foehn events and lake drainage events between figures (f.e. with shading, or labels) to better link figures and concepts across the text

- Figure 1: Consider adding a photo of the TRI in the foreground, and the illuminated surface in the background to make the viewing geometry more accessible to the casual reader. L:71 describes the illuminated ice cliff, 1: 78 the viewing geometry, but sensitivity of the TRI at such a shallow viewing angle will mostly be horizontal velocity component, and not vertical deflection as was the focus of studies investigating tidal displacement (Drews et al., 2021). It would be good to see a DEM of the glacier surface, so consider a contourmap on ice and rock. Shown is realistic, mean LOS velocity for each glaciated pixel, please add temporal standard-deviation for each pixel as well as avoid masking out the rocky areas.
- Figures 2 and 3: The 'LOS velocity' is the mean along a section of a flowline. How does the standard deviation around the presented time series of mean values evolve? Consider plotting Figures 2/3 side by side to make it easier for the reader when text compares similarities/differences between the seasons.
- Figure 4 and 5: a legend of the coloured boxes and their meaning would be useful. Different color to the presented anomalies. Put arrows down/up direction for intuitively. If it is uncertain do an up / down arrow or maybe a question mark.
- Figure 6: Couldn't this also be atmospheric variability? This is where showing the rocky areas will have a huge impact on the interpretation of the signal. Delineate the discussed blocks which are nicely discussed later for spatial heterogeneity. Colorbar labels need to be much bigger.
- Figure 7: comparison of satellite derived speed and TRI derived LOS velocity. Can the ITS\_Live be rotated into the TRI viewing geometry for a direct comparison? It is surprising to see larger variability in a LOS velocity component when compared to an absolute speed.
- Figure 8: Please add labels specifying down- to glacier acceleration, and vice versa. For panel B. It's a nice creative figure but it would help to have some guidance for getting the main point. Include modifying the caption, and provide shading to link events across figures.

- Figure A1, A2, A4, A5, A8, A9 are too small. Please increase in a revised manuscript
- Figure A6 would benefit from being shown side-by-side with satellite-derived shear strain rates from the mean ice-flow product, ideally rotated into line-of-sight (LOS) for a more meaningful comparison. At present, it is unclear how useful shear strain rates derived directly from LOS velocities actually are. Longitudinal, transverse, and shear strain rates are defined within a coordinate system aligned with the mean flow direction, whereas TRI-derived LOS velocities are strongly influenced by the instrument's viewing geometry and sensitivity patterns. As a result, calculating strain rates from LOS measurements alone does not yield physically interpretable quantities in the glacier flow frame and may be misleading without appropriate geometric transformations.
- Figure A7: seeing the rocky areas, and how the spatial pattern of the signal evolves through this lake-drainage event, would greatly improve confidence
- Figure request: Delineate ROIs of blocks and show the time series of their movement with associated standard deviations.

Thank you again for the time and care you've put into presenting this impressive TRI dataset. We're confident that the study will make a valuable contribution to the field, and we believe it is very well suited for *The Cryosphere*.

Sincerely,

Christian Wild.

---

## Author Comment (AC1)

**Author comment of egusphere-2025-5193:**
**High spatio-temporal velocity variations driven by water input at a Greenlandic tidewater glacier**

Armin Dachauer

January 2026

*We sincerely thank the reviewers and the scientific community for their thoughtful and constructive comments on our manuscript. We have carefully considered all comments and suggestions and we are convinced we can address all of them. Below, we provide detailed, point-by-point responses to each comment with our planned revisions. The original reviewer comments are noted below in Roman type, whereas our response is in italics.*

**1 Review comment 1: William David Harcourt**

**Summary**

The study of Dachauer et al. (2025) utilises a high temporal resolution terrestrial radar data set of a tidewater glacier in south Greenland to study its response to meltwater inputs to the glacier system and the impact on ice flow. They show that, even at the end of the summer season when the subglacial hydrological system is likely predominantly efficient, that diurnal fluctuations in ice velocity correlate to changes in air temperature, suggesting that the system remains active and sensitive to meltwater inputs. Furthermore, during periods of reduced ice velocities, the glacier sped up in response to the drainage of a lake (which I assume was an ice-marginal lake although the text was a bit unclear on this). Finally, the authors suggest that the diurnal signal propagates from the upper glacier to the terminus on slow flow days, but during velocity speed up events the diurnal signal propagates from the terminus and up glacier. The results demonstrate the sensitivity of the glacier to meltwater inputs over shorter timescales even in late summer when the subglacial system has transitioned to a predominantly efficient system.

**General Comments**

I found this paper to be an interesting read and the analysis that has been done nicely highlights the interesting dynamics associated with the glacier. Measurements such as these are hard-won - it takes significant logistical planning and often long periods in the field collecting the measurements. The authors should be congratulated on generating such a useful data set! I have provided technical comments below that ask for clarification in several areas. I have a few more substantive comments that I ask the authors to consider in their revision:

- Methods: The methods are insufficiently described in the paper. Although the TRI measurements have been widely described in previous studies, the paper still needs to include a description of the specific set up used in this study, including key equations to derive displacement measurements, derive 3D data (i.e. DEMs) and other processing steps e.g. coregistration (e.g. due to wind buffeting of the radar).
  *The authors understand the point raised by the referee and will describe the used method more thoroughly. In particular, the process used to derive the velocity maps will be described in depth.*

- The upward and downward propagation of the diurnal velocity signal is not clear to me. After looking at Figs. 4 and 5 several times I can start to see it, but I think it needs a bit of explanation. You could use an example of where the upward/downward propagation signal is clear in a single figure - you have attempted this in Fig. 6, but it would be useful to also show the 1D profiles as an example. My broader point is that these are referenced several times in the discussion and it is sometimes very hard to see it - case in point is Fig. 8 where these propagation patterns are mentioned several times but I struggle to see the patterns.
  *We see the difficulty to observe the upward and downward propagation of the diurnal velocity signal in Figs. 4 and 5, mostly due to the packed plot where the entire 2 weeks of data are shown simultaneously. To address this issue - which was also mentioned by other reviewers - we will provide an additional plot with a zoom-in example of Figs. 4c/5c accompanied by supporting labels and arrows to better visualize the upstream versus downstream propagation.*

- The implications of the study are not effectively described in the context of Greenland Ice Sheet dynamics and future changes. Part of the problem here is that a research question is not stated in the introduction, so it currently reads a bit more as a description of results and interpretation of them. These then need to be combined to provide an holistic overview of the glacier dynamics. I.e. the overarching conclusion is that the glacier is sensitive to meltwater inputs in late summer when the subglacial system should have an efficient drainage system. Quantifying the sensitivity (see comments below) would enable you to compare this to other outlet glaciers, but also whether the processes observed at EKaS are representative of other regions. Articulating a clear process-driven framework for the processes and then expanding that view to the whole of Greenland would be a nice way to summarise the data in a way that brings all the analyses and interpretations together.
  *The referee raises a fair point. We will refine the elaboration of our research question in the introduction. A comparison with other outlet glaciers will be challenging, since not many studies with comparable temporal/spatial resolution exist. Nevertheless, we will add an additional paragraph where we set our results in context of other regions/glaciers and reflect on wider implications of our results.*

**Technical Corrections (References to line (L) numbers in preprint)**

- L1-2: Aren't ice discharge and frontal ablation essentially considered the same?
  *While the two processes behind ice discharge and frontal ablation differ, it can be considered the same under the assumption of a steady-state, which is not given here. However, to avoid confusion and to make the meaning more clear, we will rephrase that sentence.*

- L5: If the data were gridded on a uniform grid, you should state the resolution precisely.
  *The grid is not uniform. The range resolution is about 0.75 m, while the azimuth resolution is*

*0.4˚, which corresponds to 6.9 m at a slant range of 1 km. This is already stated in the methods section but we will try to make this more explicit in the text there.*

- L6-7: Sensitivity in terms of rapid change in velocity in response to meltwater inputs? Or something else? Be precise.
  *Exactly. Will be stated more clearly*

- L13-14: You may touch on this later, but it would be useful to state a clear outcome of the paper succinctly - the relationship between meltwater and dynamics is complex and widely studied, so what exactly does this study find that contributes to this knowledge?
  *We will rephrase the sentence to make it clearer.*

  *The authors are thankful for the list of suggestions below for improving the introduction (comments L16 - L63 below). We will include all proposed changes in the revised manuscript.*

- L16-18: Glaciers are still the largest contributor, check out the recent GLAMBIE community estimate (link). There are also more recent references that should be cited alongside these e.g. Otosaka et al. (2024).

- L20: I wouldn't say it is limited? It's a very active research area! I would instead tease out the key processes of most interest e.g. meltwater and basal sliding feedbacks, ocean thermal forcing, precipitation changes.

- L23: E.g. studies such as Tedstone et al. (2015) (link) that showed long-term slowing down of a land-terminating glacier in response to larger melt input to the bed.

- L28: Optical and SAR are imagery, I think you mean feature speckle tracking in optical and SAR image pairs.

- L30: Far more than a few studies.

- L32-34: Remove the tidal forcing part from this sentence - use a separate sentence to discuss tidal impacts on velocities.

- L34-35: Broad statement that is not always true. I think it's important to acknowledge the complexities here e.g. drainage may shut down over winter (not always if meltwater can be stored at the bed). In spring, there may be a spike in velocity due to a sudden input of meltwater, then at the end of summer velocities may decline. Diurnal changes can be observed, whilst long-term changes are less well known due to observational constraints.

- L36: Challenges? I assume you are referring to measuring the inaccessible bed?

- L39-40: Sentence needs to be qualified with the reasons why we need short-term and high resolution observations. To observe diurnal patterns? Response to calving?

- L58-63: It would be good to highlight other GPRI / terrestrial radar studies to investigate tidewater glacier dynamics as there are now quite a few. You could dedicate a whole paragraph to this, or at least a few sentences, particularly discussing the benefits and limitations of the approaches and processing complications.

- L60: What is the research question being addressed?
  *We will refine the elaboration of our research question in the introduction.*

- L74: Could you put this into perspective with other glaciers? How does it compare to the big tidewater glaciers in Greenland e.g. Jakobshavn, Helheim etc.?
  *Will be added.*

- L84: Do you have a picture of the set up? Could add this to Figure 1. It's to confirm visually the field of view.
  *Will be added either in Fig. 1 or the Appendix.*

- L87-89: This should go in the data processing section, including a slightly expanded explanation of how the DEMs were generated and then the DEM processing.
  *Will be moved to the data processing section and expanded.*

- L99: Did they capture the regional temperature trends? Figs A1 and A2 seem to, but would be good to qualify this sentence with a statement saying that they do and any biases.
  *The visual proof of Figs. A1 and A2 will be supplemented by a more concise statement that the regional temperature trends are captured.*

- L118: No mention of how the DEMs were derived?
  *Will be mentioned in a new subsection in the data processing section.*

- L119-127: Although the method is well described in previous studies, I think you do need to discuss a bit more detail about the interferometric approach including key equations of how you related phase changes to displacement. Did you have to coregister the images e.g. due to wind buffeting of the radar?
  *We will add more details about the method used to generate the velocity maps.*

- L120-121: Reword: 'The TRI transmits from a single antenna and measures radar backscatter using two receiver antennas.'
  *Will be done.*

- L124: Important to recognise throughout the paper that the temporal resolution of your measurements is 30 minutes and NOT 1 minute, as is claimed.
  *Will be stated more clearly to avoid confusion.*

- L147: Although this section is generally well-written, it was sometimes unclear when you were discussing 2023 and 2024 data. E.g. the peak-to-peak amplitudes were stated to be 0.5 m/d, but what year? I would discuss each year in turn for clarity (within each section though, I like having the results section split into themes as you have done).
  *The text will be re-written to ensure clarity about the year throughout the entire section.*

- L168: I note here that the method to extract mélange presence / absence has not been described in the methods, but should be. Same for the detection of plumes.
  *We will add both to the methods.*

- L174: It's slightly off that these are not discussed in chronological order i.e. 2023 and then 2024.
  *Will be adjusted.*

- L185: State dates.
  *Will be done.*

- L190: Reference relevant section / figures.
  *This sentence is a transition to the following subsection. We will re-write the sentence to make this more clear.*

- L194: Isn't it an ice-marginal lake? Reference the figure where it is labelled L1 - Figure A3 I think.
  *L1 is a subglacial lake, L2 is ice-marginal indeed. We will add the Figure reference A3 and make sure the lake types L1 and L2 are mentioned clearly throughout the manuscript.*

- L195: How was this calculated?
  *This is a rough estimation knowing the subglacial lake area and the rough surface elevation change from time-lapse imagery. An additional sentence will be added for clarification.*

- L196: Is there a figure showing the lake drainage event?
  *The area of the subglacial lake drainage is highlighted in the inset map of Fig. A3. Since L1 is a subglacial lake discharge event, Sentinel-2 images provide only little evidence of the event. However, we have time-lapse imagery which nicely illustrate the timing of the event. According imagery will be added to the Appendix.*

- L198: When did the lake drain? Do you have exact dates?
  *Between July 4 and July 15, as mentioned on L196.*

- L203: Reference relevant figure. Wasn't L2 a supraglacial lake?
  *L2 is an ice-marginal lake. We will add the Figure reference A3 and make sure the lake types L1 and L2 are mentioned clearly throughout the manuscript.*

- L204: True if it's a supraglacial lake, unclear if it's an ice-marginal lake.
  *The authors are confident that the water from the ice-marginal lake discharge event L2 must have drained through the glacier, as there is no other way the water could possibly drain (see Fig. A3).*

- L212-222: As noted above, a description of the methods used to calculate these should be provided. Are volumes calculated for all calving events? Might be useful to add these to Figures 2 and 3.
  *Will be mentioned in a new subsection in the data processing section. The calving events are already added in Fig. 2d and 3d, but we will make sure to mention this more clearly.*

- L218-222: What about calving and diurnal variations - are there any signals or correlations?
  *No correlation could be found on a diurnal time-scale. A clarifying sentence will be added.*

- L230: I assume the constant difference means the diurnal signal is clear at all distances from the terminus? I'd explicitly state this.
  *Exactly. We will state that more clearly as proposed by the referee.*

- L234-242: The primary point here is that diurnal fluctuations propagate from the top of the centreline profile to the terminus on low velocity days, whereas on high velocity days it propagates from the terminus. However, in 2023 this is not clear as the green boxes appear before the high velocity days. In 2024, there does seem to be a correlation. So, based on the data available, this assertion may be partially true, but I think it's worth stating that this may only hold when the 'high velocity days' are significantly larger than the base velocity level.
  *While the authors agree with the referee that the pattern is more clear in 2024, we would still*

*argue, that the statement is also valid in 2023. However, sometimes the signal is somewhat noisy (e.g. around 2023-08-09), which is color-coded in green-orange stripes. Additionally, the downwards pointing deceleration in the evening of 2023-08-09 is likely linked to the arrival of the lake discharge L2, as highlighted in A7, which might interfere with the general pattern. Thus, we still clearly observe upstream propagation on high-velocity days and downstream propagation on low-velocity days, also in 2023. To address this issue we will adjust the text for more clarity.*

- L239: Were the orange/green boxes calculated manually? Or some other way?
  *Yes, they were calculated manually by investigating the time-series of acceleration maps. We will add an according sentence in the methods section.*

- L243-252: Can you label to the side of the maps which represent downward propagating and upward propagating diurnal velocity variations. This section is a bit hard to read because you reference velocity speed up events (which relate back to Figs 2 and 3) but you showing acceleration maps, so it is not clear where I should be looking. You also mention the 'outlines' of the block like pattern of flow, but this should be labelled for clarity - I assume it relates to the step-like pattern clearly visible in Fig. 6a.
  *We will add a title/label to the panes of Fig. 6 to clarify the propagation direction. Exactly, the outlines refer to the step-like pattern most obvious on Fig. 6a. We will consider adding thin outline-lines in the map or describe more thoroughly.*

- L255-261: I would start with a comparison to other Greenlandic outlet glaciers and then consider other glaciated regions (e.g. Alaska).
  *The order of the sentences will be changed as suggested.*

- L273-275: Does the July 23 precipitation have any relationship to the velocity increase on July 25th?
  *This is a valid observation and one we also considered. However, given the generally rapid response of velocity to temperature changes (on the order of 4 hours), we consider a direct influence on July 25th unlikely, particularly given the low precipitation rate. The effect of rainfall is more plausibly indirect, acting through reduced temperatures, which likely led to a diminished diurnal velocity peak on July 24. The pronounced peak observed on July 25 remains partly unexplained; however, we know that it is strongly influenced by an acceleration at the front (see Fig. 8). We will make it more clearly that the rainfall event only influences the temperature/velocity on July 24.*

- L276-282: This is quite vague - how would wind speed affect the velocity variations? For cloud cover, I suspect it is not a straight 1-1 relationship, does your RH data provide any clues as to the relationship?
  *The sentence will be rephrased to make it more clear.*

- L302-304: I must admit, I am struggling to see the acceleration. Is it the multi-day slowdown that is important here?
  *The authors mean the acceleration phase at the upper part of the centerline in the night of Aug 10, 2023, which is best seen in Fig. 4c. However, we agree that it is not very easy to spot and we will adjust the manuscript to better highlight the matter.*

- L310-315: If the water is being channelised and propagating along the bed, would there be spatial velocity variations - i.e. high flow over the channelised region, slow flow over the less non-channelised sections (assuming no water here, or maybe there will be some form of distributed

drainage system?). I guess I am looking to see here a bit more underpinning theory related to the observation rather than simply stating water flow down and then the ice flow increased.
*The authors want to emphasize that we purposely focus on our observations and try to avoid speculative conclusions. Nevertheless, we will try to rephrase the section and better link our argumentation to existing theory.*

- L322: To prove this point, you need a graph over the same period showing freshwater discharge - I assume you more broadly mean 'melt' i.e. a temperature graph would suffice?
  *The referee is right assuming that we mean "during times of high temperature in summer", indirectly assuming that this leads to increased melt and freshwater discharge. We will adjust the sentence accordingly and refer to other work that shows increased temperatures and thus melt during the summer in Greenland.*

- L326-327: You might discuss the below, but this part is interesting - stored meltwater at the bed over winter? Possibly, considering the recent findings of Hansen et al. (2025).
  *The reference will be included and the sentence adjusted accordingly.*

- L344-345: Is it possible to calculate average diurnal melt rates and therefore volumes that are transferred to bed? This might allow you to quantify the sensitivity of the bed to extra meltwater inputs by comparing with the velocity changes over the same time period - I would expect to see a declining sensitivity over summer. Ah, I see you have done in Appendix C! But can you use this to quantify the sensitivity of the drainage system over the velocity time series in Figure 7?
  *All reviewers commented on Fig. 7 with diverging opinions (from further expanding investigation to putting it into the Appendix). The authors agree with some reviewers that it is important to discuss our observations in the context of seasonal variations and therefore, we decided to keep the figure in the main manuscript. The suggested sensitivity analysis will be challenging to assess. However, we will try to investigate the velocity sensitivity by combining the diurnal variations in velocity to those in temperature/ablation for different times of the year.*

- L376-388: It is difficult to relate the text to areas on the figure, probably because the changes are subtle and/or embedded within the multiple lines shown in Fig. 8b. Can you help to identify these locations more clearly e.g. lines on the graph, annotations? This would certainly help improve the clarity of reading in this section.
  *We will add date-periods for low-velocity and high-velocity days to highlight to which parts of the graph the given numbers belong. Additionally, colors (e.g. yellow-ish for terminus) will be appointed to the text to further clarify.*

- L377: 'were selected'
  *Will be added.*

- L378: 'time-series extracted and shown in Fig. 8'
  *Will be adjusted.*

- L377-378: Maybe I missed it, but I am not sure the term 'upstream transition regime' has been explicitly stated as this before.
  *Will be made consistent.*

- L381-383: Not clear to me what this relates to in Fig. 8b?
  *We will add "... along the centerline (i.e. all lines overlap and show a similar magnitude), with values..." to clarify.*

**Figures**

- Figure 4: Might be more intuitive to have the brighter colours representing higher velocities.
  *We applied the reverted color-scale for both Figure 1 and Figures 4/5, which we want to have consistent. We came to the conclusion, that the current color-scale is easier to read. In particular, the higher velocity values towards the front are easier distinguishable with stronger colors (Fig.1) and also the diurnal cycles are better visible that way (Figs. 4/5).*

- Figure 8: Can you add a color scale indicating the distance from the terminus?
  *Distance information will be added to the graph.*

**2  Review comment 2: anonymous referee**

**Summary**

This manuscript investigates short-term ice-flow speed variability at Eqalorutsit Kangilliit Sermiat, South Greenland, over sub-diurnal to multi-day timescales using Terrestrial Radar Interferometer (TRI) measurements. The authors document clear diurnal and multi-day velocity variations that are associated with episodic water inputs, including surface melt, supraglacial lake drainage, and subglacial or marginal lake drainage events. A significant difference between the two transition regimes is the glacier's response at the terminus, where acceleration and deceleration during high-velocity periods are larger than at locations further upstream. When water input decreases, basal water pressure appears to collapse in a block-wise pattern, resulting in abrupt deceleration of basal sliding at the terminus that subsequently propagates upstream. During such events, this mechanism overrides the downward-propagating diurnal signal observed during low-velocity periods and prevents the more uniform acceleration and deceleration behavior of the glacier.

The study demonstrates the strong potential of TRI observations to resolve short-term velocity changes, and the resulting dataset is highly valuable. Overall, the interpretations are well aligned with the scope of The Cryosphere and provide meaningful insights into short-term glacier dynamics. I have some suggestions to improve this manuscript, as below.

**General Comments**

- The advantages of using TRI are not sufficiently articulated. In particular, the rationale for the chosen temporal resolution (30 minutes) and cut-off period (3 hours) is unclear. Please provide justification for these choices, including discussion of coherence, signal-to-noise considerations, and any thresholds applied.
  *We will state the justification for the 30 min resolution and smoothing period in the manuscript. Additionally, we will provide information about the thresholds we used in regards to data quality (signal-to-noise).*

- L58-63: This is not the first study to use TRI for this type of analysis. Additional relevant literature should be cited (e.g., Drews et al., 2021; Holland et al., 2016; Voytenko et al., 2015; Xie et al., 2018, 2019).
  *The study will be better embedded into existing TRI investigations.*

- L77-86: The TRI azimuth resolution varies with range. If the region of interest is within 6 km, what is the effective resolution near the ice front and at upstream locations? While the resolution

of 6.9 m at 1 km and the maximum system range of 16 km are stated as system specifications, more detailed information is needed for the actual region of interest. Please also specify the multilook factors applied and explain why these choices were made.

*More details about the azimuth resolution at the region of interest and processing parameters will be added.*

- L104: The plume extents are classified as (1) none, (2) small, (3) medium, and (4) large. Please clarify the criteria or thresholds used for this classification. Was this done qualitatively (e.g., visual inspection) or quantitatively? A similar clarification is needed for L168 regarding the definitions of small, medium, and large mélange-covered areas.

  *The classification was done qualitatively through visual inspection of time-lapse imagery from the opposing hill. We will clarify in the text accordingly.*

- L236: The statement describing a lag of approximately two hours between the terminus and the location 6 km upstream is unclear. For example, deceleration appears to occur earlier and more strongly at the terminus on 2023-08-10, 2024-07-15, and 2024-07-20. Please clarify how the lag was determined and how it varies among events.

  *The statement highlights the average time-lag of the acceleration sign change between the front and upstream regions. This can either be earlier on the front on high-velocity days or earlier upstream on low-velocity days. However, the actual time-difference fluctuates from day to day, usually between 1-3 hours, but 2 hours on average. We will make this point more clear in the text.*

- L243-252: The description of Figure 6 would benefit from clearer visual guidance. Please consider annotating the figure (e.g., with arrows or highlighted regions) to indicate the areas discussed in the text, as the current presentation is not very intuitive.

  *We are thankful for the feedback and will adjust the graph with visual support from labels and/or highlighted regions.*

- L262: The manuscript states that a Fourier analysis of the two velocity time series revealed strong 24 h peaks for both years, but the results are not shown. To support the subsequent conclusion that this reflects a solar rather than lunar influence (12.4 h), the Fourier analysis should be presented, at least in the Appendix.

  *An according figure will be added in the Appendix.*

- L274: The manuscript suggests that the relatively low precipitation rate resulted in only a small velocity response. Please clarify the basis for this interpretation. Is this because precipitation does not immediately translate into enhanced basal lubrication? Is there any relationship between precipitation and the acceleration observed on 2024-07-24?

  *We assume the referee refers to the acceleration observed on 2024-07-25. This is a valid observation and one we also considered. However, given the generally rapid response of velocity to temperature changes (on the order of 4 hours), we consider a direct influence on July 25th unlikely, particularly given the low precipitation rate. The effect of rainfall is more plausibly indirect, acting through reduced temperatures, which likely led to a diminished diurnal velocity peak on July 24. The pronounced peak observed on July 25 remains partly unexplained; however, we know that it is strongly influenced by an acceleration at the front (see Fig. 8). We will make it more clearly that the rainfall event only influences the temperature/velocity on July 24.*

- L298: If Sentinel-1 imagery is not shown or analyzed in this paper, please clarify why it is mentioned here, or include the corresponding results in the Appendix.
  *The Sentinel-1 imagery will be provided in the Appendix.*

- L303: The manuscript refers to acceleration during the night of 2023-08-10. However, this pattern is not clearly shown in Figures 2a and 4c. Please clarify this interpretation or revise the text and/or figures accordingly.
  *The acceleration phase on the upper part of the centerline in the night of Aug 10, 2023 is best seen in Fig. 4c. However, we agree that it is not very easy to spot and we will adjust the manuscript to better highlight the matter.*

- L369: Several previous studies have used GPRI for similar analyses, rather than relying solely on GPS point measurements along the glacier centerline. These studies should be cited, including Kane et al. (2020), Xie et al. (2018, 2019), and Drews et al. (2021).
  *We will set our study in context of previous studies that used TRI data.*

**Figures**

In general, the figures, axis labels, tick labels, colorbars, and legend texts are too small and difficult to read. Please enlarge them to improve clarity.
*We be done.*

- Figure 7: This figure appears to add limited new information to the manuscript. Consider moving it to the appendix.
  *All reviewers commented on Fig. 7 with diverging opinions (from further expanding investigation to putting it into the Appendix). The authors agree with other reviewers that it is important to discuss our observations in the context of seasonal variations and therefore, we decided to keep the figure in the main manuscript, but further discuss its implication (see reviewer 1).*

- Figure A3: Please consider also showing the Sentinel-2 images corresponding to drainage event L1.
  *Since L1 is a subglacial lake discharge event, Sentinel-2 images provide only little evidence of the event. However, we have time-lapse imagery which nicely illustrate the timing of the event. According imagery will be added to the Appendix.*

- Figure A4 and A5: Including 2D spatial velocity maps as animations or videos would be more effective in illustrating diurnal velocity fluctuations.
  *The presentation of Figs. 4 and 5 were mentioned by several referees with different propositions. To best account for all suggestions, we will provide an additional plot with an example zoom-in accompanied by according labels and arrows (see suggestion by reviewer 1).*

- Figure A7: Please consider overlaying the glacier centerline to improve interpretability, particularly in panel (a).
  *Will be done.*

**Minor comments**

- The manuscript uses both "ocean-terminating glacier" and "marine-terminating glacier." Please choose one term consistently throughout the text (or, "tidewater glacier").
  *We will make sure to be more consistent.*

- L69: The statement that the glacier's terminus has advanced over the past few decades should be quantified (e.g., by how much).
  *According numbers and potentially a reference to another paper in press will be added.*

- L73: Citation error (also occurs at L330).
  *Thank you for pointing out.*

- L125: The statement "which has a sensitivity of less than 1 mm" requires a reference.
  *Will be provided.*

- L206: Please consider showing the Sentinel-1 data, at least in the Appendix.
  *Will be provided.*

- L237: "Fig. 4c and Fig. 4c" typo.
  *Thank you for pointing out.*

- L287: "Fig. 5d" no such panel exists.
  *Thank you for pointing out.*

**3 Community comment 1: Christian Wild**

Dear Armin Dachauer and co-authors,

My name is Christian Wild, and I am a geophysicist at the University of Innsbruck, Austria. I have experience working with terrestrial radar interferometry (TRI) in both polar and alpine environments and have recently discussed Dachauer et al. in my MSc-level literature seminar. Our group's discussion raised a number of stimulating points that we believe could contribute to further strengthening the presented conclusions. We outline our comments below and hope that our feedback will be useful to the authors.

**Summary:**

The study investigates short-term velocity variations (sub-diurnal to multi-day) at the terminus of the tidewater glacier Eqalorutsit Kangilliit Sermiat in South Greenland. Using a terrestrial radar interferometer (TRI) with high spatial (metres) and very high temporal (1 minute) resolution, the authors capture line-of-sight velocity changes and relate them to hydrological forcing. Sub-diurnal velocity variability is associated with surface melt, and multi-day speed-up events are linked to lake drainage. Important conclusions are made towards the mechanistic link between structural controls (rift systems) modulating hydrologically induced flow.

This dataset is exceptionally rare, and we acknowledge the considerable logistical and environmental challenges required to collect it. At the same time, the richness of the observations implies that additional analyses could help test the robustness of the proposed mechanisms and extract even more insight. Our comments are offered with the intention of helping the authors make the most of the already-processed data and the code they have available, without the need to constrain an ice-dynamical model with their observations to support some of their conclusions.

**Strengths:**

- The novel resolution provided by the TRI enables unprecedented detail of glacier dynamics, which are outside the resolution of conventional satellite or UAV campaigns. Its richness offers substantial potential for further scientific insight.

- The study reveals that the glacier's response to hydrological forcing is more complex than a simple uniform sliding mechanism, with contrasting behaviour observed up-glacier versus down-glacier. The possibility that existing rift networks modulate these responses is both important and intriguing, and represents a meaningful scientific contribution.

- TRI data processing is inherently complex and often difficult to convey in a clear and accessible manner. The authors demonstrate that they mastered the method, and present the workflow in a way that is both understandable and reproducible. The figures are clear and thoughtfully constructed. In particular, the creative Figure 8 is especially effective in illustrating and synthesizing the key ideas discussed, and deserves special acknowledgement for how well it supports the interpretation.

**Weaknesses:**

- In its present form, the paper reads as observational rather than improving our understanding on a system-level or the broader context. The paper could connect subglacial hydrology more directly to implications for mass loss, calving behaviour and interactions with ice melange, plumes versus terminal velocity (Helheim Glacier, Melton et al., 2022) or even sea-level projections. One possible approach, with the processed data at hand, is to look along ice-cliff transects (or even better delineating the rift clusters, a.k.a. blocks for more detailed analysis of the proposed mechanisms). There is a figure buried in the SI, to investigate the proposed mechanistic link without the need for modeling experiments. Figures 2 and 3 are getting there, but are then disregarded in the text in this context.

- The TRI data set is not fully explored. The advantage of the system is its high spatio-temporal resolution (as reflected in the title), but temporal downsampling from the TRI's native resolution of 1 min to 3 h, and lack of spatial heterogeneity of the presented analysis (flow-parallel transects, versus possible 2d fields) raises doubt about the robustness of the results. Further analysis of shorter/longer temporal baselines, especially the evolution of rocky areas and coherence, would greatly strengthen the conclusions about the subglacial hydrology, by excluding atmospheric variability, or spatially heterogeneous ablation, as alternative mechanisms captured by the system.

**Major comments:**

1. Why are rocky areas excluded from the analysis? The TRI illuminates not only the moving ice surface, but also relatively stagnant rock faces and slopes. These areas can be used to assess the level of background noise, but even more importantly also to clearly distinguish between real glacier acceleration and apparent motion due to atmospheric variability. If the detected accelerations of the ice surface (up/down glacier and vice versa) also occur on the rocky areas nearby, clearly they are driven by atmospheric variability, if not, the author's reasoning in Section 3.2.2 are strengthened. Mean velocity fields (Fig. 1) and mean centerline velocity averages (Figs.

2 and 3, averaged over the first 2 km, but excluding a 100 m section right at the terminus. The justification for using only this part remains rather vague. See Minor comment 1) are presented, but not their standard deviations. How does LOS velocity variability change over time and space? Is there a connection to range when looking at the mean standard deviation of each pixel? How does velocity variability compare to the determined velocity increase of $15 - 30\%$ above average speed?

*The authors agree with the referee and the editor - who already raised the same point - that velocity time-series on rock faces nicely illustrate the uncertainty of the instrument, while at the same time highlighting the atmospheric variability and its influence on the data quality. Therefore, we will provide an according rock velocity time-series in the Appendix. Additionally we will provide variability measures (e.g. standard deviation) to make sure an uncertainty assessment is provided.*

2. Temporal downsampling. How robust are the results when shorter or longer temporal baselines are used to calculate interferograms? How was the presented temporal baseline of 30 min chosen in the first place? We understand that TRI data were acquired at 1 min intervals, stacked to 30 min and then smoothed using a 3 h low-pass filter to derive the presented values, which are then correlated with the time series of air temperature and relative humidity. Wouldn't it make more sense to similarly smooth the AWS data for a better comparison to the TRI-derived numbers? Moreover, the propagation speed of radar waves is controlled by the absolute amount of water vapour in the air (Goldstein 1995) so using relative humidity is not an ideal quantity in this regard, we suggest to use absolute humidity or specific humidity, as typically used to study adiabatic processes in atmospheric science. Other processing choices of the TRI were assumed to be well established, but they are not yet. How were these determined? In particular, can you provide reasoning for your choice in the temporal baseline, stacking interval for averaging, filtering, and geocoding.

*We will state the justification for the temporal downsampling (i.e. 30 min resolution) and smoothing period in the manuscript (see comment reviewer 2). The purpose for showing the relative humidity is the evidence of a foehn event around 2024-07-20, characterized by very low relative humidity and a change in wind direction, rather than the influence on data quality. To test the data quality we will show a graph with the velocity time-series on a rock surface. Other TRI velocity processing parameters will be provided in the methods (see comment reviewer 1 and 2).*

3. Some major conclusions read deterministic (e.g., L416 'driven by additional water input from surface melt', or L422 'could clearly be linked to lake drainage events') without direct observations in the subglacial system, such as ground-penetrating radar or additional evidence from modeling experiments. Does the temporal coverage of the TRI data set justify these conclusions, if only the end of a drainage event is captured? In contrast, these core conclusions are only made from 'weak' temporal correlation (L158: 0.6 at a lag of 4 hours) without rigorous analysis of their statistical significance. We suggest moving these conclusions, which are only partly warranted by the data analysis, to a dedicated discussion section, if no further evidence from auxiliary analysis are presented.

*Given the strong correlation between air temperature and velocity for periods with not additional signal such as lake discharge events (0.7 in 2023 and 0.8 in 2024), together with the fact that the timing of L1 and L2 fits well with the periods of low cross-correlation, we are confident that the given conclusion is justified. However, we will rephrase this sentence to better clarify the evidence for our conclusion.*

4. Given the spatial detail resolved by the TRI measurements, we would appreciate a more in-depth examination of how the rift system mechanically influences hydrologically driven flow near the terminus. Previous studies (e.g., Ultee et al., 2022) generally assume spatial homogeneity when assessing terminus variability, but here we have the opportunity to investigate this signal at much finer spatio-temporal scales. We therefore recommend focusing the analysis on ROIs where individual blocks exhibit distinct behavior, and we request an additional figure showing the locations of these blocks and their respective LOS velocity evolution over time.

   *The temporal investigation of individual blocks is indeed very interesting. Having this motivation in mind, the authors tested several option to best visualize the processes. Finally, we decided that the spatial detail can best be presented with a combination of Fig. 6 and Fig. 8. At Fig. 8, different colours belong to points within the individual "blocks", and therefore show the LOS velocity evolution over time for different sections at the glacier (see section 4.6).*

**Minor comments:**

1. TRI data processing: How was the geocoding performed? What DEM? Temporal baseline, how was this determined to be a representative time scale? What pixel was used for the phase unwrapping, and how was it chosen (coherence?)? What is the temporal variability of the phase at the chosen pixel and how does that add to the overall uncertainty? L134: 'Where data quality is highest' reads subjective, is there a threshold in coherence applied? L82 please add or reword to mentioning range/azimuthal resolution at the actual study site, which is $3 - 9$ km away.

   *The temporal baseline was discussed in major comment 2, the uncertainty in major comment 1. We will further elaborate the geocoding, phase unwrapping and the data quality threshold. The azimuth resolution at the area of interest will be added (see reviewer 2).*

2. General level of processing: How was the plume size determined? Was there a quantitative component to the use of "small", "medium" or "large"? L87: Were calving events detected manually, semi-automatically or automatically from the TRI derived DEMs? Similar for time-lapse imagery (L104), how was ice melange extent and subglacial plume extent quantified? L106: Add when the subglacial lake drainage event occurred approximately to help the reader.

   *The specific method of the plume, ice melange and calving extraction will be added (see Technical corrections reviewer 1). Calving events are extracted automatically from the TRI derived DEMs, the according method will be added in the data processing section (see review 1).*

3. L114: Interpretation of tides, how well is the measured tide captured by a tide model such as Greenland 1 km Tide Model (Gr1kmTM, Howard and Padman, 2021)?

   *A second pressure sensor was installed in close proximity and yielded comparable results, thereby supporting the reliability of the measurements. As the tide gauge plays only a minor role in this manuscript—being used primarily to determine the timing of diurnal tide cycles and spring tides, which we consider to be sufficiently constrained by the existing setup—we did not incorporate additional applications or products, such as the referenced tide model.*

4. L158: Are the following correlations statistically significant? How about any correlations to absolute/specific humidity?

   *The correlation to humidity values will be checked.*

5. L178: Could this be a low-pass filtered humidity signal? A comparison to a nearby rocky area, and its apparent velocity variability, might further support the observation of a real response of

the ice to melt triggered by the drastically high air temperatures.
*This issue is discussed in major comment 1.*

6. L225: Why is the analysis constrained to a centerline, and not the full spatial resolution of the TRI utilized and explored?
*The visualisation of a 2D velocity field over time is generally rather challenging. Therefore, the authors decided to provide a combination of figures using a centerline (e.g. Figs 2,3,4,5) and figures showing 2D maps of the velocity (e.g. Figs. 1 and 6). To further investigate the spatial variability, we created a plot with several flowlines (Fig. A4) and crosslines (Fig. A5), which indicate that the diurnal variability is spread homogenously throughout the entire area (see section 3.2.1). With Fig. 6 we provide a 2D velocity map to highlight the spatial variations in acceleration. Like this, we hope to present our results in the best way while still keeping the plots readable.*

7. L226: 'typical' always needs at least one e.g. type citation. Similar to the word 'generally' elsewhere.
*Citation will be added.*

8. L237: 'Fig. 4C' typo.
*Thanks for pointing out.*

9. L238: How do rocky areas behave during these times? Are they showing similar oscillations as observed on the glacier?
*See major comment 1 above.*

10. L255: Quantify 7% of mean velocity and compare it to the uncertainty of the TRI measurement.
*See major comment 1 above.*

11. L287 why cite Fig. 5c and d for foehn events? Double check the figure references. There is no panel 5d.
*Thanks for pointing out this typo. This will be changed to Fig. 3c. 5d will be removed.*

12. L327: Agree that the mean velocity fits well, but the TRI variability is much larger than the satellite-derived variability at this time of the year. Given that the TRI measures only the LOS component, one would expect a smaller mean and smaller variability when compared to the satellite? Rotating the satellite derived velocity into LOS would be beneficial to this comparison, which requires geocoding of the TRI.
*The authors agree that the TRI "only" measures LOS velocity and therefore misses a part of the absolute signal as compared to the satellite-derived velocity values. However, the set-up of the TRI is ideal in a way that it captures the large majority of the velocity signal, despite the LOS restriction. Furthermore, the purpose of the graph is to highlight the annual velocity variability of the glacier. We will adjust the figure caption/manuscript to make it more clear.*

13. L371: The sub-daily pattern could also be atmospheric variability on a daily cycle, couldn't it? Again, a comparison to nearby rocky areas would help to strengthen the observed response of the glacier.
*This issue is discussed in major comment 1.*

**Figures**

- General: Label foehn events and lake drainage events between figures (f.e. with shading, or labels) to better link figures and concepts across the text.
  *The lake drainage events L1 and L2 are already added in Fig 2 and 3, respectively. There is only one foehn event occurring during the two field periods, which is widely discussed in section 3.1.2. Since Figs. 2 and 3 are already quiet dense, we chose not to add additional shading for this single foehn event in order to avoid overloading the figures and to maintain readability.*

- Figure 1: Consider adding a photo of the TRI in the foreground, and the illuminated surface in the background to make the viewing geometry more accessible to the casual reader. L:71 describes the illuminated ice cliff, l: 78 the viewing geometry, but sensitivity of the TRI at such a shallow viewing angle will mostly be horizontal velocity component, and not vertical deflection as was the focus of studies investigating tidal displacement (Drews et al., 2021). It would be good to see a DEM of the glacier surface, so consider a contourmap on ice and rock. Shown is realistic, mean LOS velocity for each glaciated pixel, please add temporal standard-deviation for each pixel as well as avoid masking out the rocky areas.
  *A photo of the TRI set-up will be added to the Appendix (see technical corrections reviewer 1). We will check adding a contourmap as well as removing the masking and see if the readability is still guaranteed. A temporal standard-deviation will only reflect the high temporal variability of velocity during the measurement period and therefore not add any uncertainty information. However, a measure for data uncertainty will be provided (see major comment 1).*

- Figures 2 and 3: The 'LOS velocity' is the mean along a section of a flowline. How does the standard deviation around the presented time series of mean values evolve? Consider plotting Figures 2/3 side by side to make it easier for the reader when text compares similarities/differences between the seasons.
  *A measure for data uncertainty along the time-series will be provided (see major comment 1). The plotting of the two Figures next to each other was considered by the authors. Unfortunately, it comes at the disadvantage of shrinking the already well-loaded plots and reducing readability of the diurnal variability within the time-series, which are a key finding of this study. Therefore, we will keep the two plots individually.*

- Figure 4 and 5: a legend of the coloured boxes and their meaning would be useful. Different color to the presented anomalies. Put arrows down/up direction for intuitively. If it is uncertain do an up / down arrow or maybe a question mark.
  *The presentation of Figs. 4 and 5 were mentioned by several referees with different propositions. To best account for all suggestions, we will provide an additional plot with an example zoom-in accompanied by according labels and arrows (see reviewer 1 and 2).*

- Figure 6: Couldn't this also be atmospheric variability? This is where showing the rocky areas will have a huge impact on the interpretation of the signal. Delineate the discussed blocks which are nicely discussed later for spatial heterogeneity. Colorbar labels need to be much bigger.
  *Atmospheric impact was discussed in major comment 1. The blocks will be better highlighted (see referee 1). Colorbar label will be increased.*

- Figure 7: comparison of satellite derived speed and TRI derived LOS velocity. Can the ITS_Live be rotated into the TRI viewing geometry for a direct comparison? It is surprising to see larger

variability in a LOS velocity component when compared to an absolute speed.

*The authors agree that the TRI "only" measures LOS velocity and therefore misses a part of the absolute signal as compared to the satellite-derived velocity values. However, the set-up of the TRI is ideal in a way that it captures the large majority of the velocity signal. The authors are not surprised by the larger short-term variability observed in the LOS velocity component compared to the absolute speed. Short-term velocity increases (e.g. during foehn events) cannot be resolved by satellite-based velocity estimates, which provide observations only every 1–3 days, which have been smoothed by multi-day averaging necessitated by limited data availability. We consider this effect being much larger than the "signal loss" when only capturing the LOS component.*

- Figure 8: Please add labels specifying down- to glacier acceleration, and vice versa. For panel B. It's a nice creative figure but it would help to have some guidance for getting the main point. Include modifying the caption, and provide shading to link events across figures.
  *Figure will be adjusted and caption modified for more clarity.*

- Figure A1, A2, A4, A5, A8, A9 are too small. Please increase in a revised manuscript.
  *Figure sizes will be increased.*

- Figure A6 would benefit from being shown side-by-side with satellite-derived shear strain rates from the mean ice-flow product, ideally rotated into line-of-sight (LOS) for a more meaningful comparison. At present, it is unclear how useful shear strain rates derived directly from LOS velocities actually are. Longitudinal, transverse, and shear strain rates are defined within a coordinate system aligned with the mean flow direction, whereas TRI-derived LOS velocities are strongly influenced by the instrument's viewing geometry and sensitivity patterns. As a result, calculating strain rates from LOS measurements alone does not yield physically interpretable quantities in the glacier flow frame and may be misleading without appropriate geometric transformations.
  *The purpose of the LOS shear strain rate map is to highlight the position of the main rift systems, which potentially influence the block-wise acceleration and deceleration pattern. Since the absolute values and thus the uncertainty regarding the viewing geometry are secondary, the authors decided to only show the figure in the Appendix. However, to address the issue we will consider adding thin outline-lines of the rift 'blocks' in the map of Fig. 6 (see referee 1).*

- Figure A7: seeing the rocky areas, and how the spatial pattern of the signal evolves through this lake-drainage event, would greatly improve confidence.
  *Velocity time-series on a rocky area will be shown throughout the entire period, not only the lake-drainage event (see major comment 1).*

- Figure request: Delineate ROIs of blocks and show the time series of their movement with associated standard deviations.
  *The temporal investigation of individual blocks is indeed very interesting. Having this motivation in mind, the authors tested several option to best visualize the processes. At Fig. 8, different colours belong to points within the individual "blocks", and therefore show the LOS velocity evolution over time for different sections at the glacier (see section 4.6).*

Thank you again for the time and care you've put into presenting this impressive TRI dataset. We're confident that the study will make a valuable contribution to the field, and we believe it is very well suited for The Cryosphere.

Sincerely, Christian Wild.